# AgentHard: Hardening LLM-Agent Evaluation with a Taxonomy of Artifacts and Automated Cleaning

## Abstract

Reliable evaluation of LLM-based agents is often confounded by artifacts that conflate model errors with benchmark flaws, thereby misrepresenting the agents' true capabilities. To address this, we present a component-wise taxonomy of common benchmark pitfalls spanning the user, environment, evaluation, and ground truth elements of agent tasks. This analysis exposes pervasive issues such as incorrect ground-truth action sequences, ambiguous tool APIs, user simulation faults, and brittle evaluation metrics. Guided by these insights, we develop AgentBench-Cleaner, an automated pipeline in which the first two stages filter out flawed tasks: first, rule-based detectors catch deterministic errors; second, an LLM-as-a-judge identifies nuanced issues; and third, a secondary difficulty-based curation step enhances evaluation rigor. Applying the issue-filtering stages yields an issue-cleaned benchmark that removes pervasive artifacts and supports more trustworthy evaluation. The difficulty-based curation step produces a harder derivative, AgentHard-Bench, with standardized evaluation protocols and explicit quality criteria. Across diverse LLM agents, evaluations on AgentHard-Bench deliver more stable model rankings, clearer performance separations, and improved benchmark diversity relative to the original benchmarks. We will release AgentHard-Bench, along with the taxonomy and pipeline upon acceptance, to support robust, reproducible agent evaluation.

## 1 Introduction

Agent benchmarks have become essential infrastructure for evaluating and deploying large language model (LLM) agents in realistic settings. However, evaluating LLM agents remains challenging due to the complexity of their interactive tasks. In particular, unlike static single-turn evaluations, agent benchmarks require an LLM to engage in multi-turn interactions, invoke tools or APIs, and operate in dynamic environments (Zhou et al., 2023; Xie et al., 2024; Yao et al., 2024). These settings target practically useful capabilities—especially *function calling*, which bridges an agent's reasoning to concrete actions. Small choices in tool schemas or argument semantics can produce large swings in measured performance (Patil et al., 2025; Wang et al., 2025; Saha et al., 2024). Because these benchmarks guide research and deployment decisions, they must deliver reliable and discriminative assessments of agent capability.

High-quality agent benchmarks are difficult to design because tasks couple four components—*user simulation, environment, evaluation harness, and ground-truth action sequences*. This coupling introduces failure modes absent in static QA: brittle string-match evaluation, unrealistic user behavior, schema ambiguities, environment drift, and leaky assumptions across components. Recent audits have surfaced many of these pitfalls, including erroneous ground-truth trajectories, abstention-friendly tasks (where doing nothing can pass), ambiguous APIs, and user-simulation failures (Zhu et al., 2025). Such flaws permit shortcuts that inflate scores, degrade model separability, and destabilize leaderboards.

A series of agent benchmarks have pushed beyond static QA toward interactive, tool-augmented settings, including realistic web/OS environments (WebArena, OSWorld) and dialog-based tool use (MINT, $\tau$-Bench) (Zhou et al., 2023; Xie et al., 2024; Wang et al., 2024; Yao et al., 2024). Complementary function-calling evaluations (BFCL, ComplexFuncBench) tighten API semantics and

execution-based checking (Patil et al., 2025; Zhong et al., 2025). In parallel, LLM-as-a-judge methods and mixture/filter pipelines scale judgments and stabilize rankings (e.g., MT-Bench/Arena, MixEval, SMART) and offer checklists and auto-curation workflows (ABC, Arena-Hard) (Zheng et al., 2023; Ni et al., 2024; Gupta et al., 2025; Zhu et al., 2025; Li et al., 2025a). *Yet these efforts emphasize environment realism, API strictness, or rank stability in isolation, and stopping short of a unified, component-wise diagnosis of errors (User, Environment, Evaluation, Ground Truth) and an automated issue-focused per-task filtering mechanism.* We provide that missing layer: a fine-grained taxonomy of agent-benchmark issues and AgentBenchCleaner, which encodes the taxonomy into scalable detectors to filter issue-bearing tasks and yields an issue-cleaned benchmark that improves evaluation reliability. A secondary difficulty-based curation further produces AgentHard-Bench, a compact yet challenging suite with higher model separability and more stable rankings. In this work, our primary focus is the taxonomy and the issue-filtering pipeline that removes systematic benchmark artifacts; the harder AgentHard-Bench variant arises from a secondary difficulty-based curation. Our main contributions are summarized as follows:

- **Taxonomy of Agent Benchmark Issues:** We present a systematic, component-wise taxonomy of fundamental issues in LLM agent benchmarks, derived from expert analysis of representative diverse benchmarks. This taxonomy reveals common failure modes (e.g., function-call ambiguities, brittle evaluations, unrealistic user simulations) and provides a blueprint for diagnosing and avoiding such pitfalls.

- **Automated Benchmark Cleaning Pipeline:** We develop AgentBenchCleaner, an automated filtering pipeline that leverages the above taxonomy to filter out flawed tasks. It combines rule-based issue detectors with LLM-as-a-judge evaluations (augmented by selective human review) to scalably remove problematic benchmark items, constituting the core of our issue-filtering pipeline and greatly improving evaluation robustness.

- **High-Quality Benchmark Suite:** We develop an issue-cleaned benchmark composed of cleaned tasks. A secondary difficulty-based curation step yields AgentHard-Bench, a consolidated and more challenging variant that provides clearer downstream evaluation—evidenced by higher model separability and more stable model rankings compared to the existing benchmarks. Hence, AgentHard-Bench will enable clearer comparison of LLM agent capabilities and promotes more trustworthy evaluation.

## 2 RELATED WORKS

**Agent Evaluation Benchmarks.** Early LLM-agent benchmarks move beyond static QA to interactive, multi-turn settings with tool use and dynamic environments (Zhou et al., 2023; Xie et al., 2024; Wang et al., 2024; Yao et al., 2024). WebArena and OSWorld test agents on realistic web and OS tasks with automated correctness checks, while suites like MINT and $\tau$-Bench simulate dialog-based tool use in closed interaction loops. Specialized benchmarks expand coverage: ACEBench categorizes tool-use into basic, ambiguous, and multi-agent dialogue scenarios (Chen et al., 2025), and AgentBench spans domains from web navigation to code editing (Liu et al., 2023). These efforts advance realism and breadth, but their tightly coupled components expose reliability flaws—e.g., inconsistent user simulations and overly lenient success metrics ($\tau$-Bench even counted empty outputs as "successful" (Zhu et al., 2025)). These highlight the need for a more structured evaluation design.

**Function-Calling Evaluation and Tool Use.** Tool APIs are central to agent behavior, motivating benchmarks that test function-calling. BFCL evaluates correctness across diverse schemas (Python, JavaScript, SQL, REST) and patterns (sequential, parallel), executing calls to verify results (Patil et al., 2025), while CFB targets long-horizon tool use with multistep calls over 128K-token contexts (Zhong et al., 2025). They enforce strict instruction-following but assume error-free tasks and ground-truth trajectories. In practice,

Table 1: Summary of key design features of six widely used agent benchmarks.

| Benchmark | Capacity | User | Environment | Evaluation |
|---|---|---|---|---|
| **ACEBench** | Diverse tool use | Predefined, LLM | Stateful | Tool-call, final state, LLM |
| **BFCL V3** | Multi-step, Multi-turn | Predefined | Stateful | Tool-call |
| **CFB** | Complex tool call | Predefined | Stateless | Tool-call |
| **$\tau$-Bench** | Policy following | LLM | Stateful | Final state, substring |
| **$\tau^2$-Bench** | Policy following | LLM | Stateful, dual-control | Final state |
| **DrafterBench** | Policy following | Predefined | Stateless | Tool-call |

schema ambiguities and flawed "expected" calls often mislead models. We address this by filtering such tasks to ensure evaluation reflects execution semantics.

**Benchmark Filtering Pipelines.** Another line of work improves benchmarks by mixing datasets and filtering noise or overly easy items. MixEval combines existing tasks (including user queries) to yield stable rankings aligned with human-driven Arena results (Ni et al., 2024), while SMART filtering removes easy or contaminated items—shrinking datasets by up to about 70% yet improving correlation with human judgments (Gupta et al., 2025). ABC provides a high-level checklist for identifying conceptual flaws in benchmark design (Zhu et al., 2025). Its criteria (e.g., verifying that a task avoids random-guess shortcuts) are intended for human auditors and are not directly automatable at task granularity. In contrast, our taxonomy is component-aligned and task-level: each issue type corresponds to a concrete, operationalizable failure mode in the User, Environment, Evaluation System, or Ground Truth components. This enables scalable automated detection of flawed tasks rather than conceptual, design-level auditing. Thus, ABC and our approach are complementary: ABC supports human-oriented benchmark review, whereas our taxonomy is designed for automated issue filtering. Our work fills the remaining gap by providing a unified, fine-grained taxonomy of structural agent-task issues and operationalizing it into a two-stage automated issue-filtering pipeline. A lightweight, optional difficulty-based curation step then produces a harder variant for frontier-model stress testing.

## 3 SYSTEMATIC ANALYSIS OF AGENT BENCHMARK ISSUES

### 3.1 OVERVIEW OF AGENT BENCHMARKS

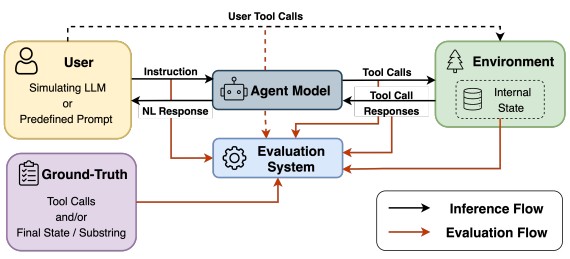

Figure 1: Illustrating generalized components of agentic AI benchmarks and their interactions.

We analyze six widely used LLM-agent benchmarks that span diverse settings and evaluation styles (see Table 1): *BFCL V3* (Patil et al., 2025), *ACEBench* (Chen et al., 2025), *DrafterBench* (Li et al., 2025c), *τ-Bench* (Yao et al., 2024), *τ²-Bench* (Barres et al., 2025), and *CFB* (Zhong et al., 2025). They differ along four structural components that together define an agent evaluation setting (see Figure 1): **User** (e.g., predefined vs. LLM-simulated, single vs. multi-turn), **Environment** (e.g., stateless vs. stateful; tool/API availability and semantics), **Evaluation** (e.g., final-state checks, tool-call matching, LLM-based judging), and **Ground Truth** (e.g., full trajectories vs. milestone steps; policy constraints). This four-component decomposition exposes failure modes that static QA does not encounter and motivates a systematic taxonomy.

### 3.2 COMPONENT-ALIGNED ISSUE TAXONOMY

We characterize and categorize recurrent benchmark issues by the component in which they originate, as shown in Table 3. Such a component-wise view turns scattered anecdotes into actionable categories and directly informs the modular detectors used in our pipeline (see Sec. 4).

**User-related issues.** User-side issues often stem from underspecified prompts that force agents to produce a single "correct" response despite open-ended instructions (e.g., some ACEBENCH and CFB tasks). In settings with *LLM-simulated users* (e.g., τ-Bench and τ²-Bench) (Yao et al., 2024; Barres et al., 2025), we observe *role confusion* where the user model produces assistant-like confirmations (e.g., "your reservation has been canceled"), corrupting dialogue flow and making agent behavior hard to judge.

Table 2: A concrete example: issue breakdown for τ-Bench.

| Component | Share (%) |
|---|---|
| User | 21.0 |
| Environment | 30.6 |
| Evaluation System | 22.6 |
| Ground Truth | 25.8 |

**Environment-related issues.** These arise when the actions available to the agent or the feedback it receives are inaccurate, misleading, or insufficient.

Table 3: A summary of the identified issue taxonomy of agent benchmarks.

| Benchmark Component | Issue Category | Description | Affected Benchmarks |
|---|---|---|---|
| User | Ambiguous instruction | The predefined user prompt is underspecified and allows multiple interpretations while the benchmark expects one specific task completion trajectory. | ACEBench, CFB |
| | User role confusion | The user simulator sends messages or behaves like an assistant rather than a user. | $\tau$-Bench, $\tau^2$-Bench |
| Environment | Incorrect tool-call responses | A tool returns inaccurate or irrelevant results that prevent the agent from completing the task correctly. | CFB, $\tau$-Bench, $\tau^2$-Bench |
| | Insufficient toolset | The environment does not provide the necessary tools for the agent to fulfill the user's request. | ACEBench, BFCL V3 |
| | Misleading tool design | Tool names or descriptions misrepresent their actual behavior. | ACEBench, $\tau$-Bench |
| | Incorrect system prompt | The system prompt itself contains errors or misleading examples that guide the agent toward invalid calls. | DrafterBench |
| Evaluation System | Too lenient | Evaluation criteria allow trivial or incomplete solutions to pass. | $\tau$-Bench |
| | Too strict | Evaluation criteria unfairly penalize semantically correct answers for minor deviations. | ACEBench, CFB |
| Ground Truth | Malformed tool calls | Ground-truth calls violate the function schema by using wrong types, invalid values, or missing arguments. | ACEBench, BFCL V3, CFB |
| | Incorrect tool calls | Ground-truth calls select the wrong function or parameters, contradicting the user's request or context. | ACEBench, CFB, BFCL V3, $\tau$-Bench, $\tau^2$-Bench |
| | Redundant/ungrounded tool calls | Ground-truth call sequences contain tool calls that are unnecessary or ungrounded by the context, causing unfair evaluation. | CFB |

(i) *Incorrect tool-call responses* provide wrong or irrelevant results even for correct queries, blocking task completion (e.g., CFB, $\tau$-Bench, and $\tau^2$-Bench). (ii) *Insufficient toolsets* omit necessary tools, rendering tasks unsolvable by construction (e.g., ACEBENCH and BFCLv3). (iii) *Misleading tool design* (names/descriptions that contradict actual behavior) steers agents toward suboptimal functions (e.g., ACEBENCH and $\tau$-Bench). (iv) *Incorrect system prompts* can hardwire invalid behavior—for example, a prompt instructing agents to call Python methods without parentheses (DRAFTERBENCH) produces systematically invalid calls.

**Evaluation-System Issues.** Evaluation criteria can be miscalibrated. Overly lenient scoring allows agents to exploit loopholes — for example, about 38% of $\tau$-Bench tasks pass if the database remains unchanged, enabling a "do nothing" strategy (Zhu et al., 2025). Conversely, overly strict criteria can reject semantically correct outputs due to brittle exact-match requirements.

**Ground-truth issues.** Errors in the benchmark's *answer key* are especially harmful because they redefine correctness. We observe: (i) *Malformed tool calls* in the reference trajectories that violate schemas (wrong types/enums, missing required arguments), penalizing agents that adhere to the API. (ii) *Incorrect function or parameters* in ground truth that contradict user intent or policy (canceling a non-cancelable item), forcing agents to mimic mistakes to receive credit. (iii) *Redundant or ungrounded steps* that add unnecessary actions; efficient solutions are marked wrong for not reproducing superfluous calls.

**Discussion.** The aforementioned issues arise across all four components rather than being concentrated in one place. For example, in $\tau$-Bench (see Table 2), the shares are: User - 21.0%, Environment - 30.6%, Evaluation - 22.6%, and Ground Truth - 25.8%. This spread motivates a component-wise design of detectors; in Sec. 4, we operationalize the taxonomy into modular rules and LLM-judge checks in our AgentBenchCleaner.

## 4 FROM TAXONOMY TO AUTOMATED FILTERING

### 4.1 OVERVIEW

Our issue taxonomy provides a principled framework for identifying and categorizing recurring issues in agent benchmarks. It reveals that many flaws are not one-off quirks of specific tasks,

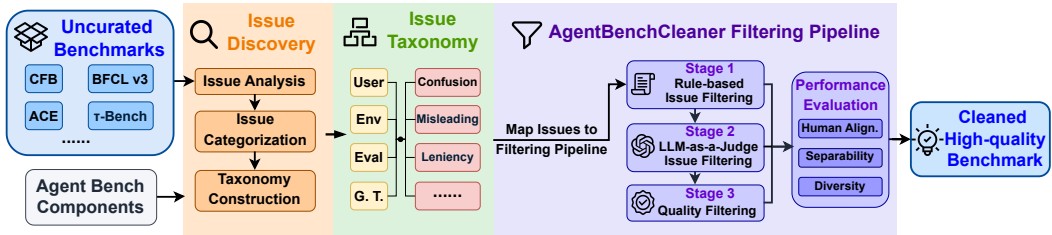

Figure 2: Overview of the end-to-end process of utilizing issue taxonomy, AgentBenchCleaner pipeline, and the final cleaned output AgentHard-Bench

but stem from systematic patterns across benchmark components and their interactions. This insight underpins a filtering approach that avoids ad hoc fixes and instead generalizes across diverse settings.

While one could theoretically resolve issues by manually annotating each task according to the taxonomy, such an expert-driven process would be prohibitively costly and unscalable. To address this, we develop AgentBenchCleaner, an automated multi-stage pipeline that operationalizes the taxonomy into a systematic and scalable method for pruning flawed tasks as illustrated in Figure 2. The taxonomy's categories serve as actionable criteria that we instantiate into filtering mechanisms, effectively extending expert judgment across large and evolving benchmark suites. In the following, we describe the pipeline's three stages, which progressively filter out problematic tasks and refine the benchmark. Stages 1 and 2 perform taxonomy-guided issue filtering, which is the core of our cleaning pipeline.

## 4.2 THE AGENTBENCHCLEANER PIPELINE

Our pipeline consists of three sequential stages. These stages work in concert to realize taxonomy-guided cleaning: a rule-based filter, an LLM-as-a-judge filter, and a final quality filter.

**Stage 1: Rule-based Issue Filtering.** The first stage targets issues that can be detected through explicit patterns or deterministic criteria. Serving as a fast first-pass, it removes tasks with obvious flaws and thus reduces the burden on subsequent stages. We derive a set of filtering rules for this stage directly from the taxonomy, focusing on categories with clear, unambiguous signals. For example, a task with a ground-truth function-call sequence that contradicts its specification can be automatically flagged, as can tasks with ambiguous API schemas (e.g., missing or conflicting parameter definitions). By enforcing these rules, we eliminate easily detectable defects upfront, allowing later stages to concentrate on subtler, context-dependent issues.

**Stage 2: LLM-as-a-Judge Issue Filtering.** The second stage addresses more nuanced issues that require semantic understanding and cannot be caught by simple rules. Here, we leverage LLMs as judges, guided by the taxonomy to evaluate each task for complex flaws. We craft prompts that instruct an LLM to check for specific issue categories in the context of the task, providing relevant details such as the ground-truth action sequence, tool/API definitions, and any user simulator behavior. Each prompt follows a general template (see Appendix A.4) informed by the taxonomy but is tailored to the benchmark and issue at hand. The taxonomy serves as a flexible guideline rather than a rigid script: it ensures systematic coverage of known issue types, while the prompting framework remains modular and extensible to new issues as they emerge. This LLM-as-a-judge stage effectively scales expert assessment and, together with Stage 1, forms the backbone of our issue-filtering pipeline allowing subtle flaws to be identified at scale that would be impractical to enumerate with hard-coded rules.

**Stage 3: Difficulty-based Filtering.** The final stage performs a secondary, difficulty-based curation of the benchmark to enhance its evaluative usefulness after obvious issues have been removed. Inspired by recent mixture-and-filter pipelines in the literature (Gupta et al., 2025), we apply a simple heuristic: filtering out tasks that nearly all models can solve. Removing these overly easy tasks prevents benchmark saturation and ensures that the remaining benchmark remains challenging, informative, and better suited for evaluating model capabilities.

### 4.3 AGENTHARD-BENCH

Applying the issue-filtering stages to the six representative benchmarks introduced in Sec. 3 yields an issue-cleaned benchmark: a curated collection of agent tasks that have been rigorously cleaned according to the taxonomy. Applying the secondary quality-filtering step then produces AgentHard-Bench, a harder derivative of this suite. This issue-cleaned benchmark serves as a high-quality benchmark for evaluating LLM agents, free from the most prominent pitfalls identified by our taxonomy. In Sec. 5, we report statistics on how many tasks are removed and demonstrate improvements in key metrics (e.g., increased model separability and diversity), with the harder AgentHard-Bench variant showing clearer comparative signals across models. We will release both the issue-cleaned benchmark and AgentHard-Bench to facilitate more reliable and informative agent evaluation, providing the community with a foundation for more trustworthy comparisons and future benchmark development.

## 5 EXPERIMENTS AND RESULTS

### 5.1 EXPERIMENTAL SETUP

**Evaluation metrics.** To validate the effectiveness of the AgentBenchCleaner pipeline, we conduct experiments focusing on two objectives: (1) measuring human alignment to assess the accuracy of issue detection, and (2) quantifying improvements in benchmark quality metrics such as separability, diversity, and compression rate. For human alignment, we adopt two complementary protocols to measure consistency with expert judgments:

- *Balanced subset validation:* human experts annotate 10% of tasks (with a minimum of 30 tasks) using controlled sampling to construct a balanced evaluation set with a 50:50 ratio of issue and non-issue tasks. This approach ensures sufficient representation of both classes and enables reliable precision and recall estimation for the LLM-as-a-judge filtering stage.

- *Post-hoc validation on the full benchmark:* after running the full pipeline, human experts manually verify all tasks identified as issues by the pipeline across the entire benchmark to measure false positives and true positives at scale.

Rule-based filtering is deterministic by construction and thus achieves perfect alignment. Thus, human validation primarily targets the LLM-as-a-judge filtering stage.

For benchmark quality, we evaluate three key metrics: separability, diversity, and compression. Separability is quantified using standard metrics such as model agreement rates (Gupta et al., 2025) and separability with confidence (Li et al., 2025b). To fairly contextualize separability with confidence (e.g., CI non-overlap), we compare it against a baseline obtained by randomly sampling tasks of the same size. Diversity is measured through embedding-based distance metrics, where we embed task prompts using the Qwen3-Embedding-8B model and compute pairwise cosine distances. Compression is defined as the percentage reduction in task count after filtering, reflecting the extent of flawed or saturated task removal.

**Implementation details.** We use Gemini-2.5-pro-thinking (Google, 2025) as the default LLM-as-a-judge model, chosen for its strong reasoning capabilities and robustness in structured evaluation tasks. Prompt templates for this stage are provided in Appendix A.4. The rule-based filtering stage is validated automatically using predefined criteria derived from the issue taxonomy. We constructed a leaderboard and evaluated 16 LLMs on the six benchmarks, including a mix of proprietary and open-source systems that are representative of diverse model families as reported in Table 5.

### 5.2 MAIN RESULTS: AGENTBENCHCLEANER VALIDATION

We report results validating the effectiveness of the AgentBenchCleaner across six representative agent benchmarks. The results focus on evaluating the pipeline's main motivation of scaling human efforts in issue detection, with additional analyses examining how the secondary difficulty-based curation stage improves downstream benchmark utility. To this end, we present results on two primary aspects: human alignment and quality metric improvement of AgentBenchCleaner.

Table 4: Human alignment and benchmark quality metrics before and after applying the Agent-BenchCleaner pipeline.

| Benchmark | Human Alignment | | | Separability | | | |
|---|---|---|---|---|---|---|---|
| | Precision | Recall | F1-Score | Model agreement ($\downarrow$)[1] | CI non-overlap ($\uparrow$)[2] | Diversity ($\uparrow$)[3] | Compression ratio |
| ACEBench | 0.846 | 0.917 | 0.880 | 0.735 (**-0.138**) | 0.236 (**+0.181**) | 0.506 (**+0.013**) | 68.0% |
| BFCL V3 | 0.805 | 0.805 | 0.805 | 0.620 (**-0.034**) | 0.817 (**+0.025**) | 0.332 (**+0.001**) | 23.1% |
| CFB | 0.875 | 0.840 | 0.857 | 0.572 (**-0.040**) | 0.825 (**+0.025**) | 0.492 (**-0.005**) | 23.4% |
| $\tau$-Bench | 0.786 | 0.733 | 0.759 | 0.617 (**-0.040**) | 0.467 (**+0.059**) | 0.157 (**-0.043**) | 33.3% |
| $\tau^2$-Bench | 0.867 | 0.867 | 0.867 | 0.658 (**-0.016**) | 0.592 (**+0.050**) | 0.235 (**-0.015**) | 39.6% |
| DrafterBench[1] | - | - | - | 0.791 (**-0.021**) | 0.642 (**+0.094**) | 0.278 (**-0.014**) | 82.9% |

*Notes.* Values in parentheses indicate the difference between before and after applying AgentBenchCleaner. [1] *DrafterBench:* all issues are detected by rule-based filtering. [2] Model agreement: change relative to the initial value. [3] CI non-overlap: change relative to the randomly sampled baseline. [4] Diversity: change relative to the initial value.

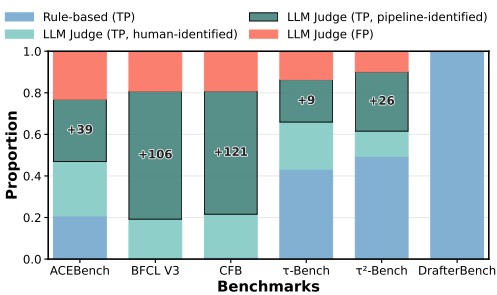

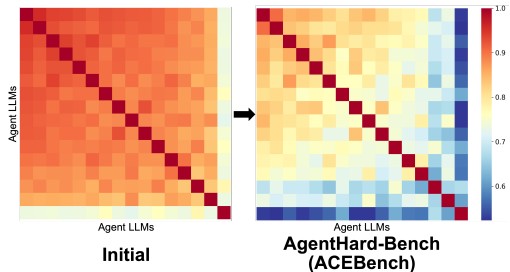

Figure 3: *Post-hoc* breakdown of tasks flagged as issues by the AgentBenchCleaner pipeline.

Figure 4: Improved model agreement for ACEBench before and after applying Agent-BenchCleaner.

### 5.2.1 HUMAN ALIGNMENT OF AGENTBENCHCLEANER

We first evaluate the pipeline's alignment with human expert judgments using the balanced subset validation. Table 4 reports precision, recall, and F1 scores across all benchmarks, excluding DrafterBench where all issues are detected by rule-based filtering. The results show consistently strong alignment, with F1 scores ranging from 0.759 to 0.880, indicating that the LLM-as-a-judge filtering stage effectively captures nuanced issues identified by experts.

To assess scalability beyond the sampled subset, we further conduct *post-hoc* validation on the full benchmarks. Figure 3 presents the manual verification results against the tasks flagged by the pipeline. The results indicate that the pipeline maintains high accuracy at scale, with end-to-end accuracy at least about 77 %. Moreover, the pipeline identifies a substantial number of previously undetected issues, with up to 121 newly discovered cases in CFB. These findings confirm that Agent-BenchCleaner not only aligns closely with human judgments but also generalizes beyond the labeled subset, effectively scaling expert-level evaluation to large benchmarks.

### 5.2.2 BENCHMARK QUALITY IMPROVEMENTS

We next evaluate benchmark quality before and after applying the complete AgentBenchCleaner pipeline, including the secondary difficulty-based curation stage. As shown in Table 4, the full pipeline consistently improves benchmark quality across all key metrics. Separability shows notable gains, with model agreement rates decreasing by an average of 0.0482 and confidence interval (CI) non-overlap increasing by 0.072, indicating clearer differentiation between model capabilities. As illustrated in Figure 4, a heatmap visualization further highlights this improvement, showing reduced agreement among models and sharper distinctions in their performance. The pipeline does not substantially reduce diversity, as the embedding distance remains largely stable, showing that the process effectively preserves a broad range of task types. Finally, the full pipeline yields an average compression ratio of 45.0% across benchmarks, reflecting a substantial reduction in task count while retaining challenging and informative tasks. Together, these results demonstrate that Agent-

Table 5: Performances on $\tau$-Bench across the initial dataset and the versions after issue filtering and after the full pipeline (*AgentHard-Bench*). Parentheses show rankings relative to the initial dataset. Cells highlighted in blue indicate models with ranking changes at each step.

| Num. | Model Name | Initial | Issue-Filtered | AgentHard-Bench ($\tau$-Bench) |
|---|---|---|---|---|
| 1 | O3-high | 0.685 (1) | 0.711 (2) | 0.652 (2) |
| 2 | Claude-4-opus-thinking-off | 0.667 (2) | 0.719 (1) | 0.697 (1) |
| 3 | Claude-4-sonnet-thinking-on-10k | 0.667 (3) | 0.686 (3) | 0.629 (4) |
| 4 | GPT-4.1 | 0.642 (4) | 0.636 (7) | 0.573 (7) |
| 5 | O4-mini-high | 0.636 (5) | 0.645 (6) | 0.596 (6) |
| 6 | DeepSeek-V3.1-thinking-off | 0.624 (6) | 0.669 (4) | 0.618 (5) |
| 7 | Kimi-K2-Instruct | 0.624 (7) | 0.669 (5) | 0.640 (3) |
| 8 | GPT-4o-20240806 | 0.594 (8) | 0.603 (9) | 0.573 (8) |
| 9 | Claude-4-sonnet-thinking-off | 0.588 (9) | 0.579 (10) | 0.528 (11) |
| 10 | DeepSeek-V3-0324 | 0.582 (10) | 0.620 (8) | 0.551 (9) |
| 11 | Qwen3-235B-A22B-Thinking-2507-FP8 | 0.558 (11) | 0.562 (11) | 0.539 (10) |
| 12 | GPT-4.1-mini | 0.479 (12) | 0.512 (12) | 0.461 (12) |
| 13 | Qwen3-235B-A22B-FP8 | 0.455 (13) | 0.471 (13) | 0.449 (13) |
| 14 | GPT-4o-mini | 0.436 (14) | 0.463 (14) | 0.382 (15) |
| 15 | Qwen3-235B-A22B-Instruct-2507-FP8 | 0.406 (15) | 0.430 (15) | 0.404 (14) |
| 16 | GPT-4.1-nano | 0.194 (16) | 0.174 (16) | 0.146 (16) |

BenchCleaner not only scales issue detection but also produces a harder derivative of benchmarks that offers clearer comparative signals, maintains diversity, and efficient for evaluating LLM agents.

### 5.2.3 PRACTICAL IMPACT: LEADERBOARD SHIFTS

To further demonstrate the practical impact of the AgentBenchCleaner pipeline, we analyze how model leaderboards and performance gaps change before and after filtering. Reliable benchmarks should produce stable rankings that accurately reflect model capabilities while maintaining sufficiently large performance gaps to ensure meaningful differentiation between models. We therefore examine how filtering affects both leaderboard positions and performance disparities across LLMs. In particular, we investigate how removing benchmark issues influences ranking stability and measured performance, and we present the resulting changes after applying the complete end-to-end pipeline to deliver AgentHard-Bench.

Table 5 compares the model leaderboard scores for $\tau$-Bench across the initial version, the issue-filtered version, and the final AgentBenchCleaner pipeline. The results show substantial shifts: final rankings changed for 75% of the models, with an average shift of 1.12 positions. Notably, the top two positions exchanged places, and their performance gap widened from 0.018 to 0.045. We also observe a resolved tie between Claude-4-opus-thinking-off and Claude-4-sonnet-thinking-on-10k, where a previously tied score now exhibits a clear performance difference. These findings underscore the importance of benchmark quality for accurate evaluation and highlight the value of the AgentBenchCleaner pipeline in producing a more reliable, informative, and practically useful evaluation infrastructure. Additional analysis of leaderboard ranking changes, including a bump-chart visualization of ranking movements corresponding to Table 5, is provided in Appendix A.5.

### 5.3 AGENTHARD-BENCH STATISTICS

We provide a comprehensive summary of the AgentHard-Bench, detailing the number of tasks retained and removed at each stage of the pipeline across the six benchmarks. Figure 5 presents these statistics, highlighting the effectiveness of each filtering stage. The issue filtering stage removes an average of 32.1% of tasks, while the difficulty-based curation stage further eliminates 35.4%, resulting in a final compression rate of 56.1%. To prevent the complete removal of any task category, we add a safeguard to retain at least 10% of tasks from each category. In addition to aggregate compression

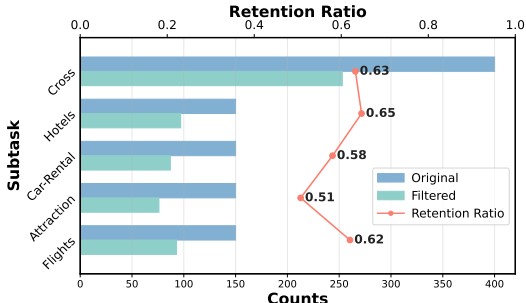

Figure 5: Retention ratio of each subtask category in CFB before and after applying AgentBench-Cleaner

Table 6: Robustness of judge models for the LLM-as-a-judge issue filtering stage.

| Benchmark | Performance | Judge Models | | |
|---|---|---|---|---|
| | | DeepSeek-V3.1-thinking-on | Gemini-2.5-pro-thinking-on | Claude-4-opus-thinking-on-10k |
| ACEBench | Precision | 0.781 | 0.846 | 0.778 |
| | Recall | 0.694 | 0.917 | 0.778 |
| BFCL V3 | Precision | 0.745 | 0.805 | 0.917 |
| | Recall | 0.950 | 0.805 | 0.550 |
| CFB | Precision | 0.742 | 0.875 | 0.825 |
| | Recall | 0.920 | 0.840 | 0.660 |
| $\tau$-bench | Precision | 0.750 | 0.786 | 1.000 |
| | Recall | 0.800 | 0.733 | 0.400 |
| $\tau^2$-bench | Precision | 0.722 | 0.867 | 0.786 |
| | Recall | 0.867 | 0.867 | 0.733 |

*\* Notes. DrafterBench:* all issues are detected by rule-based filtering.

statistics, we provide a qualitative analysis of subtask-level retention to verify that capability coverage is preserved after filtering in Appendix A.7.

## 5.4 ABLATION STUDIES

### 5.4.1 ROBUSTNESS OF JUDGE MODELS

To evaluate the robustness of the LLM-as-a-judge filtering stage, we compare its performance across different judge models. Because the task requires strong reasoning capabilities, we select models known for their reasoning strength, including Gemini-2.5-pro-thinking (Google, 2025), Claude-4-opus-thinking-on (Anthropic, 2025), and the open-source DeepSeek-V3.1-thinking-on (DeepSeek, 2025). Table 6 reports the human alignment results of the pipeline across six benchmarks for each model. The results show that all judge models achieve a similar level of alignment with human annotations with an average precision and recall of 0.836/0.832, 0.748/0.846, 0.861/0.624, respectively. These findings demonstrating that the pipeline remains robust to the choice of LLM judge.

### 5.4.2 STAGE-WISE ABLATIONS OF AGENTBENCHCLEANER

We conduct ablation studies to dissect the contributions of each stage in the AgentBenchCleaner pipeline. We summarize the benchmark quality metrics after each filtering stage compared to the initial benchmarks in Appendix A.6. The results show that each stage contributes meaningfully to overall improvements, with the rule-based filtering stage providing an initial reduction in flawed tasks, and the LLM-as-a-judge stage further refining the set. Difficulty-based curation enhances separability and diversity, confirming the value of each component in the pipeline.

## 5.5 CASE STUDIES

In this section, we present representative examples of issues detected by our pipeline. We begin with cases where the pipeline effectively uncovers diverse benchmark flaws, thereby extending human evaluations. We then examine failure cases in which the LLM-as-a-judge makes incorrect judgments. We first highlight two representative pipeline detections.

**User role confusion in $\tau^2$-bench**. In a $\tau^2$-bench telecom scenario, the user simulator generated the message "It looks like your phone is currently set to '2G only' " revealing a clear role-confusion error. This sample was effectively filtered out by the rule-based step, which flagged the frequent occurrence of the phrase "your phone" in user messages across task-completion trajectories.

**Incorrect tool call in BFCL V3**. A ground truth in a BFCL V3 sample calls a Unix-like `touch` command to `node.md`, although it was asked for `notes.md`. The LLM-as-a-judge flagged this case under incorrect tool call, recognizing that the parameter value contradicted the user's instruction.

Additional case studies are provided in Appendix A.1. We also present the failure modes of the judge model in Appendix A.2. Collectively, these findings reveal key limitations of automated

filtering and suggest directions for improvement, such as stronger prompting for scenarios that may possibly mislead the judge model.

To illustrate that our pipeline can also facilitate targeted task repairs when maintainers prefer fixing over filtering, we include three representative repairable cases in Appendix A.3, where the structured reasoning traces directly pinpoint the minimal edits required to correct flawed tasks.

# 6 LIMITATIONS

Our work has several limitations. First, the difficulty-based curation step is designed specifically to produce a harder variant (AgentHard-Bench) for frontier-model evaluation; consequently, its composition depends on the reference models used for stratification. To ensure long-term comparability across diverse model families, we recommend using the issue-filtered benchmark, the primary output of our pipeline. Second, AgentHard-Bench is not intended for evaluating weaker or mid-scale models, which may require the broader difficulty range preserved in the full issue-filtered set. Third, our pipeline focuses on identifying issue-bearing tasks rather than repairing them. While reliable automatic repair of complex agent trajectories remains an open challenge, the structured reasoning traces produced by our detectors can aid human-in-the-loop repair as shown in Appendix A.3, which we leave for future work. Finally, although our taxonomy is easily applicable to unseen benchmarks, it may require extension as new agentic task types and interaction modalities emerge.

# 7 CONCLUSIONS

We tackle a central obstacle in LLM-agent evaluation: benchmark artifacts that confound measured capability. Our approach couples a component-wise taxonomy of issues with an automated, scalable cleaning pipeline—AgentBenchCleaner—that combines rule-based detectors, LLM-as-a-judge checks, and a final quality curation step. Applying this process yields AgentHard-Bench, a suite that is cleaner, more discriminative, and more informative for agent assessment. Across six representative benchmarks, the pipeline is validated to align strongly with expert judgment and consistently improves benchmark quality: separability increases, diversity is maintained or improved, and the cleaned suites compress to roughly half their original size, exposing clearer performance gaps and more stable rankings. Hence, it can be expected that our work will enable the development of more effective LLM agent benchmarks and capable LLM agents.

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

# A APPENDIX

## A.1 CASE STUDIES

This appendix provides additional case studies that expand on the examples in Section 5.5.

**ACEBench.** We identify issues arising from user queries, provided toolset, and the ground truth annotation. First, some user queries are underspecified; for example, a task that asks for "some climate data" fails to specify the detail level, which is required by the tool schema. Second, the toolset can be insufficient to solve some problems and sometimes misleading: one example is `vlookup_formula_generator`, which has a parameter `exact_match` that performs an exact match when set to false. Finally, ground truth annotations are often flawed or contain malformed inputs.

**BFCL V3.** We find the issues in the available toolsets and the ground truths. First, the toolset is insufficient for certain tasks; for instance, a user requests a travel-time estimate, but only a distance-estimation tool is provided. Second, the ground truth contains various errors: malformed calls using strings instead of integers, flawed or redundant tool calls, such as using an incorrect file name or requiring unnecessary sorting before counting the characters in file system-handling tasks.

**CFB.** This benchmark contains issues regarding the integrity of the environment, evaluation system, and the ground truth. First, the environment yields incorrect tool responses, such as resolving "Melbourne" to Florida instead of Australia. The evaluation system is occasionally too strict, requiring exact string matches for coordinate values, marking tool calls that use rounded values wrong. Additionally, the ground truths are plagued by data type violation, incorrect parameter values (e.g., searching LA instead of requested NYC), and redundant steps.

**DrafterBench.** We find an error in the system prompt design. The prompt provided to the agent contains syntactically incorrect Python code examples, instructing to perform a method call without parentheses.

$\tau$**-Bench.** We observe problems in user simulation, tool descriptions, and evaluation validity. First, the user simulator exhibits role confusion, producing assistant-like messages such as "I can look up your reservation". Second, some tool definitions are misleading. An example is `search_onestop_flight` being described as searching for direct flights. Third, the evaluation is excessively lenient, allowing a trivial "do-nothing" model to achieve a 38% success rate. Lastly, the ground truth contains policy violations, such as canceling basic-economy reservations, which is explicitly prohibited by the given system policy.

$\tau^2$**-Bench.** This benchmark suffers from user role confusion, environment errors, and incorrect ground truth. Similar with $\tau$-bench, the user model occasionally experiences role confusion and the ground truth actions often violates the given policy. Additionally, we discover that the tool responses are sometimes unreliable, such as retrieving orders for a user different from the requested ID.

Table 7 summarizes all identified issues, organized by the benchmark component, their specific issue category, and representative examples drawn from each benchmark.

## A.2 CASE STUDY OF LLM-AS-A-JUDGE FAILURE MODES

In this appendix, we provide the failure cases of LLM-as-a-judge based on a comprehensive review of the evaluation results, where the model incorrectly flags an issue-free sample as flawed (false positives) or fails to identify a flawed sample (false negatives). Below are the four identified sources of false positive decisions:

- **Logical inference failure**. The judge model often fails to infer the intended ground truth using straightforward logical reasoning. For example, in a BFCL V3 sample, the user requests the agent to delete the message sent "in the previous turn," and the ground truth correctly deletes the most recently sent message. However, the LLM-judge marks the sample as flawed by pointing out that another message might have been sent outside the displayed dialog, which is an overly skeptical and unreasonable assumption.
- **Misjudgment due to partial trajectory**. The context provided to the judge omits many intermediate interactions between the user and the agent. In practice, agents may issue con-

Table 7: Comprehensive summary of benchmark issues and representative cases

| Benchmark Component | Issue Category | Benchmarks & Representative Cases |
|---|---|---|
| User | Ambiguous instruction | **ACEBench**: A user asks for "some climate data," but the function requires a detail-level field (Summary/Detailed), making the request ambiguous. |
| | User role confusion | $\tau$-**Bench**: The user model says "No worries! I can look up your reservation using your user ID." 
 $\tau^2$-**Bench**: The user model responds "Your reservation has been canceled." |
| Environment | Incorrect tool-call responses | **CFB**: The call `Search_Flight_Location` resolves the input "Melbourne" to Florida instead of Australia. 
 $\tau^2$-**Bench**: A tool retrieves orders made by `sofia_hernandez_5364` instead of `sofia_hernandez_8513` as it is asked. |
| | Insufficient toolset | **ACEBench**: The sample forces the agent to use a culturally focused landscape tool to answer a broad land-use change request. 
 **BFCL V3**: The user requests a travel-time estimate, but only a distance-estimation tool is available. |
| | Misleading tool design | **ACEBench**: The parameter `exact_match` in `vlookup_formula_generator` contradicts its actual behavior by performing an exact match when it is set to false. 
 $\tau$-**Bench**: The function `search_onestop_flight` is described as searching direct flights, which is misleading. |
| | Incorrect system prompt | **DrafterBench**: The system prompt shows Python code that calls a method without parentheses. |
| Evaluation System | Too lenient | $\tau$-**Bench**: A trivial do-nothing model succeeds in 38% of cases. |
| | Too strict | **CFB**: The evaluation system requires an exact string match for latitude/longitude, despite multi-dimensional matching being allowed elsewhere. |
| Ground Truth | Malformed tool calls | **ACEBench**: The field `schedule.time` violates the required HH:MM regex because it provides a time range ("08:00–17:00") instead of a single time. 
 **BFCL V3**: A ground-truth call `close_ticket` is invoked with the string `ticket_id` instead of an integer. 
 **CFB**: A ground-truth call provides latitude/longitude as floats instead of strings, violating the schema. |
| | Incorrect tool calls | **ACEBench**: The ground truth schedules the task with a "High" priority instead of the requested "Urgent," using an inappropriate scheduling tool. 
 **BFCL V3**: A ground-truth call creates `note.md` instead of the requested `notes.md`. 
 **CFB**: A ground truth searches for a taxi in LA, although the request was for NYC. 
 $\tau$-**Bench**: A ground truth cancels a basic-economy reservation, violating policy. 
 $\tau^2$-**Bench**: A ground truth cancels a departed flight, violating policy. |
| | Redundant/ungrounded tool calls | **BFCL V3**: The agent is asked to display last ten lines after sorting a file; The ground truth calls `sort` followed by `tail`, while `tail` call, which prints last lines of the original, unsorted file, is redundant. 
 **CFB**: The user instructs the agent to continue until it finds an attraction that meets a specified criterion, but the ground truth invokes `Get_Attraction_Details` in an arbitrary order and continues after the condition is met, producing redundant tool calls. |

firmations or follow-up questions in natural language, and some benchmarks expose only milestone tool calls rather than full sequences, none of which is included in the judge's context. Although the prompt instructs the judge to reasonably infer such unobserved interactions (Appendix A.4), it sometimes fails to do so. For example, in a CFB task requesting a trending museum, many agents correctly call a generic attraction-search tool and identify museums in their natural-language response, but the judge mislabels this as an insufficient-toolset flaw, noting the absence of a dedicated museum-finding tool.

- **Insufficient provided context**. Some misclassifications stem from the lack of necessary context about the internal state of the environment. For example, in a retail-domain sample of $\tau^2$-bench, the set of relevant order details is not trivial and thus is difficult to determine statically when constructing the prompt. This caused the judge to misclassify a few samples as using hallucinated order details.

- **Misjudgment of redundancy**. The judge model often labels required steps as redundant, marking valid samples as flawed. For example, in a BFCL V3 sample, an agent performs a login step because it is the only way to identify the currently authenticated user; However, the judge misinterprets this as unnecessary, claiming that the log-in is redundant since the account is already authenticated.

We also identify the modes of false negative decisions, based on the categorization of their actual issues. We note that no false negatives were found for issue categories not listed below, either because those categories fall outside the scope of what LLM-judge aims to detect or because such cases occur infrequently.

- **Malformed or incorrect tool calls overlooked**: The judge sometimes fails to identify tool calls that are incorrect or directly violates API specification. For example, in a BFCL V3 sample, a tool explicitly requires distance inputs in miles, yet the agent supplies values in kilometers. Despite this clear mismatch, the judge does not flag the call as incorrect.

- **Incorrect tool responses overlooked**: The judge also occasionally fail to identify incorrect or irrelevant tool responses. For example, in a CFB sample, an attraction-finding tool queried for the Tokyo city center returns results located in the suburbs, yet the judge does not flag the sample as flawed.

- **Ambiguity overlooked**: Some samples that resolve an ambiguous user instruction arbitrarily were not discovered by the LLM-judge. One instance is a CFB car-rental sample, which asks for vehicles that meet "conditions of reimbursement" without specifying the exact conditions. However, the ambiguity was not identified by the judge.

- **Insufficient toolset overlooked**: The judge occasionally misclassifies tasks as solvable even when the available tools are insufficient. For example, in a CFB sample that requires filtering only SUVs, the toolset provides neither an SUV-specific filter nor car-type information in the generic search results. Nonetheless, the judge incorrectly marks the agent's behavior as non-flawed.

We present the detailed breakdown of each failure category in Table 8.

Table 8: Breakdown of the failure modes of LLM-as-a-judge.

| Type | Category | Percentage (%) |
|------|----------|----------------|
| **FP** | Logical inference failure | 11.3 |
| | Partial trajectory | 17.7 |
| | Insufficient provided context | 1.6 |
| | Misjudgement of redundancy | 8.1 |
| **FN** | Malformed/incorrect tool-calls overlooked | 46.8 |
| | Incorrect tool responses overlooked | 8.1 |
| | Ambiguity overlooked | 1.6 |
| | Insufficient toolset overlooked | 4.8 |

### A.3 REPAIRABLE CASES ENABLED BY THE PIPELINE

While our evaluation focuses on filtering, the structured reasoning traces produced by the pipeline also naturally support task repair when benchmark maintainers prefer fixing over removal. Below we provide three representative examples where the reasoning trace directly identifies the minimal edit needed to correct a flawed task.

#### A.3.1 BFCL: GROUND-TRUTH FILENAME ERROR (GROUND-TRUTH ISSUE).

**Task:** `multi_turn_long_context_10`

**Reasoning trace:** "Turn 3: The user requests creation of a file named `notes.md`, but the ground truth calls `touch(file_name='note.md')`, which misspells the filename. Turn 4 again refers to `notes.md`, while the ground truth continues with the incorrect name. This contradicts the user's instructions in both turns."

**Minimal fix:** Replace all occurrences of `note.md` with `notes.md` in the ground-truth sequence.

#### A.3.2 ACEBENCH: API SCHEMA VIOLATION (ENVIRONMENT ISSUE).

**Task:** `normal_single_turn_single_function_50`

**Reasoning trace:**

"The user wants to create a morning routine for their son. The ground-truth function call uses the `FamilyRoutineManager_createMorningRoutine` tool, which is correct for the task. However, the parameter `startTime` is set to '07:30'. The API schema restricts `startTime` to one of {'06:00', '07:00', '08:00'}, so the call violates the tool specification."

**Minimal fix:** Adjust the argument to a valid enum (e.g., '07:00') or update the schema to allow free-form times.

### A.3.3  COMPLEXFUNCBENCH: UNDERSPECIFIED USER INSTRUCTION (USER ISSUE).

**Task:** `Car-Rentals-49`

**Reasoning trace:** "The user asks for the booking summary of all vehicles that meet 'the conditions for reimbursement', but these conditions are never defined. The ground truth arbitrarily selects three vehicles to summarize, which is unsupported by the user's request and logically inconsistent."

**Minimal fix:** Specify the reimbursement conditions explicitly in the user prompt (e.g., price threshold, mileage, or rental duration).

These examples illustrate how the pipeline's component-level reasoning traces substantially reduce the effort needed to diagnose and repair flawed tasks, complementing the primary use case of issue filtering.

### A.4 LLM-AS-A-JUDGE PROMPT TEMPLATE

We carefully designed the prompt for the LLM-as-a-judge filtration step. Although each benchmark differs in the implementation of benchmark components, we designed the judging instructions to share a common structure: (i) define the LLM's role as an expert evaluator, (ii) enumerate all inputs the judge receives (e.g., conversation history, system policies, available toolset), (iii) specify the issue categories with corresponding examples, and (iv) provide decision criteria and the required output format. The issue categories were specified according to our issue taxonomy discussed in Section 3. However, issues irrelevant to flaws in individual samples, such as those related to the evaluation system, were omitted from the prompt. Some issues, such as user role confusion, were also not specified in the prompt since they can be addressed better in rule-based filtering.

Based on this template, only benchmark-specific details were adapted. For example, for benchmarks that do not provide the full ground truth trajectory, we added a separate section that instructs the judge to actively reconstruct the unshown conversation. Additionally, issues irrelevant to each benchmark were not listed in the prompt to avoid misleading the judge model and causing false positives.

We provide the prompt used to filter out $\tau^2$-bench samples as a representative example, since it consists of all the design choices that would complicate the evaluation: an LLM-simulated user, direct tool calls from the user, non-exhaustive ground-truth tool-call trajectory, complex domain-specific policy, and persistent state involving internal database. This representative prompt illustrates the full range of instructions and issue categories used in our benchmarks; prompts for other benchmarks follow the same template with minor, benchmark-specific adjustments. We note that this template can be readily adapted to other agentic benchmarks beyond those covered in this paper.

---

**LLM-as-a-Judge Filtration Prompt for $\tau^2$-Bench**

```
You are an expert evaluator for Tau-2-Bench, a benchmark designed to assess an agent's
    ability to follow complex rules and interact with a simulated user.
Your task is to determine if a given benchmark sample has a fundamental flaw in its user
    prompt, environment, or ground-truths, which would make it unable to be incorporated
    in the evaluation.

You will be provided with the following information:
```

---

* **Task Description/Instructions**: The prompt or scenario given to the model that
  simulates user. You need to expect how the model that simulates the user would
  behave given this instruction.
* **System Policy**: Domain-specific rules that the agent model needs to obey. This will
  be given as the system prompt for agent models.
* **User Context and Relevant Information**: a brief information of the user and relevant
  information. This may be in a form of system message given to the user-simulating
  model.
* **Initial State**: The initial environment setup and conditions before the task begins.
  This defines the starting state of the system.
* **Tools available to the agent**: a list of tools available for the agents and their
  schema.
* **Tools available to the user**: a list of tools available for the user-simulating
  models and their schema. When the user cannot directly call any tools, this is set
  to be empty.
* **Complete Evaluation Criteria**: The complete evaluation criteria including milestone
  ground-truth actions (function calls), and final environment state assertion to
  validate. The sample is considered to be flawed if one or more of these criteria is
  unachievable.
* **Ground-Truth Milestone Tool Calls**: the provided ground-truth trajectory. Note that
  this is not a complete log of all tool calls. Instead, it is a curated list
  containing only the key milestone tool calls required to solve the task. Note that
  messages with `"role": "observation"` are the outputs of the immediately preceding
  tool call.
A sample is **flawed** if it exhibits one or more of the issues described below.

## Flaw Categories

Below is the categorization of benchmark issues, outlined according to its **relevant
benchmark component**. A sample is considered flawed if it has one or more of the
issues below.

### Environment

This category covers flaws within the agent's operating environmentthe tools and API
resultswhich can make a task unsolvable regardless of the agent's logic.

* Insufficient toolset: the environment does not provide the necessary tools (functions),
  making it impossible for the agent to solve the task even with a combination of
  multiple tools and reasoning.
  * Example: A user asks for an advanced file manipulation, while the environment only
    provides basic tools like `mkdir` or `ls`.

* Misleading toolset design: the naming or the description of an available tool is
  misleading or contradicts its actual functionality.
  * Example: A tool named `vt_get_votes_on_ip_address` provides "example.com" as an
    example for its argument value in its schema.

### Ground-Truth

This category addresses errors in the provided ground-truth trajectory, where the
supposed correct solution is itself incorrect, forcing any correct agent to fail the
evaluation.

* Malformed tool calls: A technical error where a ground-truth tool call violates the
  provided API schema.
  * Example: A parameter requires a string but is given a number (e.g., dest_id: 123
    instead of dest_id: "123"), a required parameter is missing, the tool name is
    wrong, or a parameter value is misspelled (e.g., sort_by: "popularitye" instead of
    "popularity").

* Incorrect tool calls: A tool call is syntactically valid but logically flawed. The tool
  choice or a parameter value contradicts the user's request or the context from
  previous steps.
  * Unjustified/hallucinated parameters: A value (e.g., a date, a coordinate) that
    appears without any grounding context. For example, searching for a hotel on a
    date that was not returned by a preceding flight search.
  * Contradictory: A value that directly contradicts a constraint in the user's prompt.
    However, it is NOT a flaw if there is any chance that the agent's action was a
    necessary alternative due to constraints like an insufficient budget or a lack of
    available seats.
  * Policy violation: A tool call in the ground truth trajectory directly violates the
    provided system policy. Example: The ground truth where the agent uses a specific
    tool twice, although it is mentioned in the system policy that the tool can only
    be used once.
  * Misspelled or incorrectly identified parameter values: A misspelled name or an
    ID/slug that points to the wrong entity (e.g., selecting the wrong airport ID).

```
  * Redundant/ungrounded tool calls: The ground truth tool call trajectory consists of tool
      calls that are redundant in solving the task, ungrounded by the context, or
      irrelevant in solving the task.
    * Irrelevant tool call: A tool call in the ground truth trajectory is totally
        irrelevant to the task or belongs to a completely different domain. Example: agent
        calls a tool to reserve a flight, though it was asked to process product exchange.
    * Redundant tool call: A tool call that is not necessary in solving the task. Example:
        the agent is asked to search for attractions until it finds one that meets a
        certain condition; However, the agent performs the search in an arbitrary order,
        resulting in an excessive number of tool calls.

## Crucial Rule: Actively Reconstruct the Conversation

The ground-truth trajectory only contains key milestone tool calls. It intentionally
    omits tool calls that are less important for evaluation and the natural language
    conversation between the user and the agent (e.g., user confirmations, request,
    clarifications, or follow-up questions).
Your task is to find undeniable flaws. Therefore, you MUST operate under the following
    assumption:

* For example, the ground truth milestone sequence may not contain a call that
    authenticates the user identity. It may have been intentionally omitted from the
    milestone sequence, since it is considered less important than calls that explicitly
    process user requests. Therefore, lack of authentication, user's confirmation or
    request, clarification should NOT be the sole reason to judge a sample as flawed.
* If a sequence of function calls can be justified by a plausible, un-shown conversation
    that does not contradict the User Scenario or System Policy, then it is NOT a flaw.
    The agent would have explained the user why it cannot process the request, although
    it is not shown in the milestone trajectory.
* In other words, imagine a possible conversation history that would justify the ground
    truth milestone tool call trajectory. When you consider a plausible trajectory, note
    that the user can make a request that is not mentioned in the prompt, guided by the
    agent. Flag a sample as flawed ONLY if a tool call is impossible to justify, even
    with a hypothetical conversation. Do NOT infer a flaw from missing conversational
    steps.

## Evaluation and Output Format
Carefully analyze the provided sample. Think step-by-step to determine if the
    ground-truth trajectory is a correct and logical solution to the user's prompt.

Your final output must be a JSON object with the following structure, with no additional
    commentary:

```json
{{
  "reasoning": "Provide a clear, step-by-step explanation for your decision. If the
      sample is flawed, specify what is incorrect and why it contradicts the user's
      prompt, system policies, or the user's role. If it is not flawed, briefly explain
      why the sample is valid.",
  "reasoning_summary": "A shorter rationale for your decision. If the sample is not
      flawed, just mention that it is not flawed. If it is flawed, specify the issue
      concisely. e.g., The ground truth books a connecting flight, but the user
      requested a direct flight.",
  "error_category": "The category that corresponds to the issue. e.g., \"Incorrect tool
      calls\". If the sample is not flawed, use \"Not Flawed\".",
  "is_flawed": <true or false>
}}
```

## Target Sample

### Task Description/Instructions

```
{instruction}
```

### System Policy

{agent_system_prompt}

### User context and relevant information

{user_context}
```

```
### Initial Status

{initial_state}

### Tools available to the agent and their schema

```json
{available_tool_list}
```

### Tools available to the user and their schema

```json
{available_user_tool_list}
```

### Complete Evaluation Criteria

```json
{evaluation_criteria}
```

### Ground-Truth Milestone Tool Calls
* Note that messages with "role": "observation" are the results of the tool call right
    before.

```json
{gt_tool_call_traj}
```
```

## A.5 ANALYSIS ON LEADERBOARD RANKING CHANGE

In this appendix, we provide an extended analysis of the leaderboard ranking changes. We employ three metrics to quantitatively measure the ranking changes:

**Ranking Change Rate**. This metric is defined as the proportion of models whose ranking positions differ between the original benchmark and the filtered benchmark.

**Average Rank Shift**. This metric measures the average change in ranking position for each model. While Ranking Change Rate only captures whether a model's ranking changes or not, without considering the extent of the change, Average Rank Shift complements it by quantify the magnitude if running movements.

**Indistinguishable NUM**. We define two models as indistinguishable if the absolute difference in their performance is less than 0.01, indicating that the benchmark cannot effectively differentiate between them. The number of indistinguishable models is then computed as the total count of such indistinguishable models. This metric reflects the discriminative ability of the benchmark across different models.

Table 9 summarizes the results. As shown in the table, after applying our pipeline, at least 37.5% of the model rankings in Agent-Bench changed compared to the initial version, and the number of indistinguishable models was consistently reduced, indicating that the discriminative ability of the benchmark has been improved.

Figure 6 presents a bump chart that directly corresponds to Table 5. It highlights how individual models shift across ranking positions under (i) the initial benchmark, (ii) the issue-filtered benchmark, and (iii) the curated *AgentHard-Bench*. This visualization makes it easier to see crossing trajectories and relative movements, especially in cases where several models undergo small but meaningful shifts.

## A.6 STAGE-WISE ABLATION STUDY OF AGENTBENCHCLEANER

We present stage-wise ablation results of AgentBenchCleaner across all benchmarks to show how each metric - model agreement, CI overlap, diversity, and compression ratio - changes at each stage of the pipeline. Tables 10-15 report the results.

Table 9: Model ranking changes compared across benchmarks for initial, issue-filtered, and AgentHard-Bench versions.

| Benchmark | Performance | Issue-Filtered vs. Initial | AgentHard-Bench vs. Initial |
|---|---|---|---|
| **ACEBench** | Ranking change rate | 56.2% | 62.5% |
| | Average rank shift | 1.00 | 1.00 |
| | Indistinguishable num. | $13 \rightarrow 11$ | $13 \rightarrow 9$ |
| **BFCL V3** | Ranking change rate | 12.5 % | 37.5% |
| | Average rank shift | 0.12 | 0.38 |
| | Indistinguishable num. | $8 \rightarrow 4$ | $8 \rightarrow 2$ |
| **CFB** | Ranking change rate | 25.0% | 37.5% |
| | Average rank shift | 0.25 | 0.38 |
| | Indistinguishable num. | $4 \rightarrow 2$ | $4 \rightarrow 4$ |
| $\tau$-**bench** | Ranking change rate | 56.2% | 75.0% |
| | Average rank shift | 0.88 | 1.12 |
| | Indistinguishable num. | $8 \rightarrow 8$ | $8 \rightarrow 2$ |
| $\tau^2$-**bench** | Ranking change rate | 37.5% | 62.5% |
| | Average rank shift | 0.50 | 0.75 |
| | Indistinguishable num. | $9 \rightarrow 8$ | $9 \rightarrow 6$ |

\* *Notes. DrafterBench:* all issues are detected by rule-based filtering.

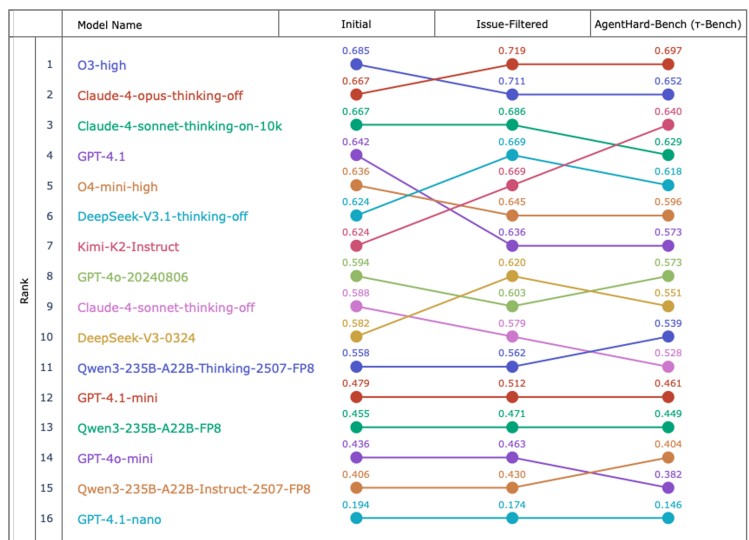

Figure 6: Bump chart visualization of model ranking changes on $\tau$-Bench across the initial dataset, the issue-filtered version, and the full pipeline (AgentHard-Bench). This is a visual counterpart to Table 5.

Table 10: Step-wise ablation on $\tau$-**Bench**

| Metrics | Initial | Stage 1 | Stage 2 | Stage 3 |
|---|---|---|---|---|
| **Agreement ($\downarrow$)** | 0.657 | 0.657 | 0.643 | 0.617 |
| **CI Overlap ($\uparrow$)[1]** | 0.475 | 0.367/0.600 | 0.392/0.317 | 0.408/0.467 |
| **Diversity ($\uparrow$)** | 0.200 | 0.185 | 0.156 | 0.157 |
| **Total Questions** | 165 | 146 | 121 | 110 |
| **Compression Ratio (%)** | 0 | 11.5 | 26.3 | 33.3 |

\* *Notes.* [1]: CI Overlap is reported as *baseline / current value*, where the baseline is a randomly sampled set of equal size.

Table 11: Step-wise ablation on **CFB**

| Metrics | Initial | Stage 1 | Stage 2 | Stage 3 |
|---|---|---|---|---|
| **Agreement** ($\downarrow$) | 0.612 | 0.612 | 0.586 | 0.572 |
| **CI Overlap** ($\uparrow$)[1] | 0.833 | 0.833/0.833 | 0.808/0.825 | 0.800/0.825 |
| **Diversity** ($\uparrow$) | 0.497 | 0.497 | 0.498 | 0.492 |
| **Total Questions** | 1000 | 1000 | 797 | 766 |
| **Compression Ratio (%)** | 0 | 0 | 20.3 | 23.4 |

\* *Notes.* [1]: CI Overlap is reported as *baseline / current value*, where the baseline is a randomly sampled set of equal size.

Table 12: Step-wise ablation on $\tau^2$-**Bench**

| Metrics | Initial | Stage 1 | Stage 2 | Stage 3 |
|---|---|---|---|---|
| **Agreement** ($\downarrow$) | 0.674 | 0.675 | 0.677 | 0.658 |
| **CI Overlap** ($\uparrow$)[1] | 0.625 | 0.533/0.550 | 0.458/0.558 | 0.542/0.592 |
| **Diversity** ($\uparrow$) | 0.250 | 0.247 | 0.239 | 0.235 |
| **Total Questions** | 273 | 228 | 179 | 165 |
| **Compression Ratio (%)** | 0 | 16.5 | 34.4 | 39.6 |

\* *Notes.* [1]: CI Overlap is reported as *baseline / current value*, where the baseline is a randomly sampled set of equal size.

Table 13: Step-wise ablation on **ACEBench**

| Metrics | Initial | Stage 1 | Stage 2 | Stage 3 |
|---|---|---|---|---|
| **Agreement** ($\downarrow$) | 0.873 | 0.869 | 0.881 | 0.735 |
| **CI Overlap** ($\uparrow$)[1] | 0.417 | 0.325/0.325 | 0.417/0.300 | 0.055/0.236 |
| **Diversity** ($\uparrow$) | 0.493 | 0.493 | 0.491 | 0.506 |
| **Total Questions** | 1023 | 996 | 901 | 327 |
| **Compression Ratio (%)** | 0 | 2.6 | 11.9 | 68.0 |

\* *Notes.* [1]: CI Overlap is reported as *baseline / current value*, where the baseline is a randomly sampled set of equal size.

Table 14: Step-wise ablation on **BFCL v3**

| Metrics | Initial | Stage 1 | Stage 2 | Stage 3 |
|---|---|---|---|---|
| **Agreement** ($\downarrow$) | 0.654 | 0.654 | 0.621 | 0.620 |
| **CI Overlap** ($\uparrow$)[1] | 0.817 | 0.817/0.817 | 0.792/0.817 | 0.792/0.817 |
| **Diversity** ($\uparrow$) | 0.331 | 0.331 | 0.332 | 0.332 |
| **Total Questions** | 800 | 800 | 618 | 615 |
| **Compression Ratio (%)** | 0 | 0 | 22.8 | 23.1 |

\* *Notes.* [1]: CI Overlap is reported as *baseline / current value*, where the baseline is a randomly sampled set of equal size.

Table 15: Step-wise ablation on **DrafterBench (Rule-based only)**

| Metrics | Initial | Stage 1 | Stage 2 | Stage 3 |
|---|---|---|---|---|
| **Agreement** ($\downarrow$) | 0.812 | 0.814 | 0.847 | 0.791 |
| **CI Overlap** ($\uparrow$)[1] | 0.842 | 0.700/0.642 | 0.667/0.625 | 0.558/0.642 |
| **Diversity** ($\uparrow$) | 0.292 | 0.288 | 0.281 | 0.278 |
| **Total Questions** | 1920 | 640 | 457 | 328 |
| **Compression Ratio (%)** | 0 | 66.7 | 76.2 | 82.9 |

\* *Notes.* [1]: CI Overlap is reported as *baseline / current value*, where the baseline is a randomly sampled set of equal size.

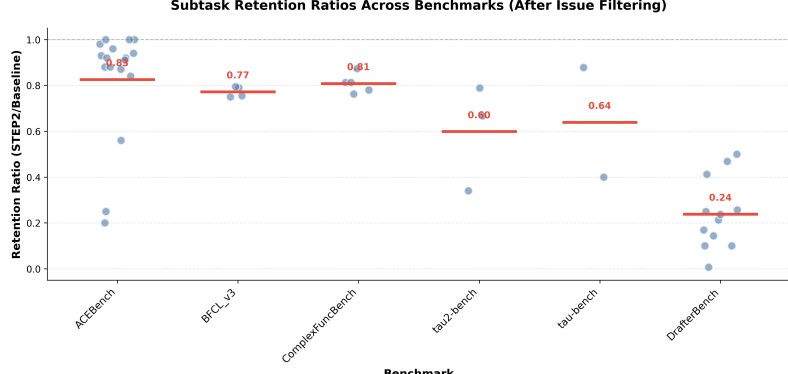

Figure 7: **Subtask retention ratios after issue filtering.** The majority of subtasks retain substantial coverage across all benchmarks, indicating that Stage 1 and Stage 2 filtering primarily removes structurally invalid tasks.

## A.7 CAPABILITY RETENTION ANALYSIS

To complement the embedding-based diversity analysis in the main text, we examine subtask-level retention and capability coverage across all benchmarks. Our goal is to verify that the AgentBench-Cleaner pipeline preserves the breadth of agentic capabilities while removing structurally flawed or saturated tasks. Figures 7 and 8 report the retention ratio after **issue filtering** and after the full pipeline including **difficulty-based curation**.

**Benchmarks with high retention and balanced capability structure.** For BFCL V3, CFBench, tau-bench, and tau2-bench, retention remains 55–77%. The remaining tasks preserve a balanced distribution of subtasks, indicating that filtering primarily removes structural issues (e.g., ground-truth inconsistencies, schema ambiguities, evaluation harness errors) rather than disproportionately affecting specific agentic skills. Thus, the issue-filtered benchmark maintains strong capability coverage while improving data validity.

**Benchmarks with benchmark-specific removal patterns.** For ACEBench and DrafterBench, retention is lower but driven by identifiable structural reasons. In DrafterBench, the `addvector` subtask is fully removed due to a verified system-prompt error that produces inconsistent ground-truth references. In ACEBench and the remaining DrafterBench subtasks, additional filtering reflects that many tasks provide limited discriminative signal for frontier models. Our pipeline includes a 10% per-subtask retention safeguard to avoid removing any capability domain entirely, ensuring that easier subtasks remain represented.

**Takeaway.** These analyses confirm that AgentHard-Bench maintains capability diversity. Filtering decisions are driven by task validity and discriminative value, not by excluding any particular skill or subtask type. The subtask-level retention ratios demonstrate that the pipeline effectively preserves the breadth of agent behaviors while removing tasks unsuitable for reliable evaluation.

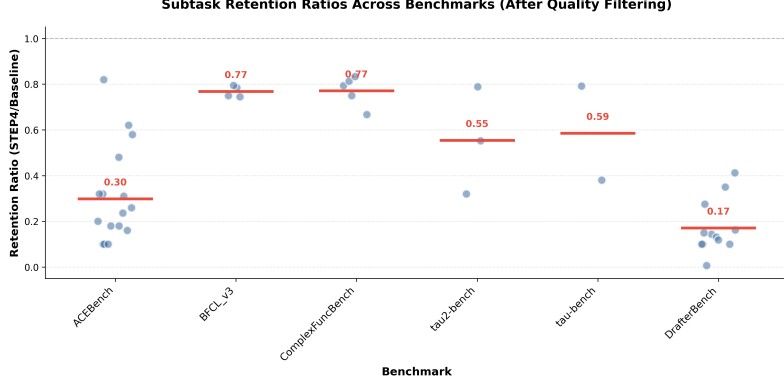

Figure 8: **Subtask retention ratios after the full pipeline (including difficulty-based curation).** Even after optional curation, a 10% per-subtask safeguard ensures capability diversity across all benchmarks.

