# OpenReview forum: "AgentHard: Hardening LLM-Agent Evaluation with a Taxonomy of Artifacts and Automated Cleaning"
_ICLR.cc/2026/Conference — Submitted to ICLR 2026_

### Official Review · Reviewer_qdgh · 2025-10-19

**Soundness:** 1
**Presentation:** 2
**Contribution:** 1
**Rating:** 2
**Confidence:** 5

**Summary:**

This paper studies LM agent benchmarks and proposes: (A) a taxonomy of benchmark failures, (B) a three-stage debugging pipeline (rule-based checks, LLM-as-a-judge, and a final "quality" filter that removes tasks that most models solve or that show low discriminative ability), and (C) trimmed versions of six existing benchmarks (compressed to $\approx 45\\%$ of original size). The intended outcome is a cleaner, more discriminative suite of tasks and improved cross-model separability (measured by non-overlapping confidence intervals).

**Strengths:**

- **Originality.**
  - The paper attempts to systematize common failure modes in LM agent benchmarks and to operationalize them into a practical debugging pipeline.
- **Quality.**
  - The authors conduct a multi-benchmark pass and surface concrete issues; several examples appear to be genuinely new (or at least under-documented) and would be useful to benchmark maintainers.
  - Automating checks via an LLM-judge could reduce maintainer overhead if validated on unseen benchmarks.

**Weaknesses:**

This paper is really problematic and I apologise for the harsh tone in advance.

1) **Circular validation akin to training on the test set.**
   The paper derives a taxonomy from the six target benchmarks, turns that taxonomy into a long, highly specific LLM prompt, and then re-applies it to the same six benchmarks. This design is training on the test set.  The Gemini LLM-judge is primed with bug patterns that are specific to the evaluation set, so its success is not strong evidence of generality.
   **Actionable fix:** Evaluate on genuinely unseen benchmarks not used in taxonomy construction, say Terminal Bench; preregister the taxonomy (frozen) and report zero-shot detection rates on new benchmarks.

2) **Unclear empirical value of the taxonomy.**
   What is the point of yet another taxonomy?  What's the point of doing this rather than the ABC (Zhu et al NIPS2025)?  Is the taxonomy useful in any way?
   It is not shown that the taxonomy increases bug-finding power beyond a simpler prompt or generic critique instructions. The taxonomy is also fairly coarse, which is tautologically exhaustive but not deeply diagnostic.
   **Actionable fix:** Provide ablations that compare (i) generic critique prompts vs. taxonomy-guided prompts, (ii) short vs. long taxonomy prompts, (iii) ABC prompt vs. prompt in this paper.

3) **Problematic pipeline design and selection-induced inflation of separability.**
   The final “quality” filter removes tasks where the frontier models can reliably solve (i.e., the easy tasks), then reports improved separability vs. randomly sub-sampled subsets of the original benchmark.  Discarding all easy tasks just to artificially inflate separation is **bad practice**.  To begin with, several benchmarks (e.g., GAIA) already come with difficulty tags, and I want the authors to explain this: why did the developers of GAIA not only keep the hard tasks?  Wouldn't that be better for the separability of their benchmark?

   Mathematically, conditioning on high variance (i.e., difficulty of the tasks) will mechanically inflate non-overlap of confidence intervals **without increasing the benchmark’s true information content.**  Consider the following setup.

   **Setup:** Let the full task set be $ \mathcal{T}=E\cup H $ with easy $E$ and hard $H$, $E\cap H=\varnothing$. For model $m$, accuracies on $E$ and $H$ are $p_m^E$ and $p_m^H$. Since the models chosen are mostly pretty strong, we're in the regime where $p_m^E\approx 1$.
   **Toy example:** Suppose $|\mathcal{T}|=100$ with $|E|=55$ and $|H|=45$, and two models differ only on $H$:
   $$
   \text{Model 1: }(p_1^E,p_1^H)=(1,5/45), \quad
   \text{Model 2: }(p_2^E,p_2^H)=(1,40/45).
   $$
   Full benchmark scores: $60\\%$ vs. $95\\%$; hard-only: $11\\%$ vs. $89\\%$. The ranking and the signal already come from $H$; removing $E$ merely rescales numbers. Comparing a “trimmed” subset enriched for $H$ to a random subset of the same size will inflate CI non-overlap by design, since we are reducing the effective number of hard tasks used in evaluation, therefore reducing the effective data size.  However, this does not mean that the trimmed down model is actually more discriminatory!  You can always only assign difficult tasks if you are trying to differentiate strong models from medium models, but throwing away easy tasks also hurts the separability for weak models versus medium models.  By the logic of the authors, if I have only weak and medium models, then I should actually discard all the hard tasks (since no medium or weak model can solve hard tasks).  Instead, I should keep only the easy tasks (**precisely the ones that authors have discarded**) and report improved separability on those models.  Is that what the authors believe should be done?

   Furthermore, in practice, if we have benchmark1 (larger, but with some easy tasks and benchmark2 (smaller, but only with hard tasks) and we want to differentiate models using the benchmarks, we would evaluate all models on the entirety of benchmark1 and benchmark2.  Being larger is precisely the core advantage of benchmark1, so why should benchmark1 be sub-sampled to the size of benchmark2?  Since we want maximal separability, it's pointless to work with a subsampled subset.

   **Actionable fix:** Re-run the experiments on a family of 16 models that are 8B or below and report if the separability of the trimmed down benchmark is still higher than the original FULL benchmark.  I find it hard to imagine that this will happen.  This upshot is that the authors provide neither sufficient justification that discarding easy tasks is a good idea, nor sufficient evidence that the trimmed down benchmark is has more separation power.

4) **Rank stability claim unsubstantiated.**
   The paper mentions improved rank stability but provides no statistical test (e.g., Kendall-$\tau$ or Spearman, with CIs under resampling) and no sensitivity to sample size.

   **Actionable fix:** Report rank correlations with 95% CIs across $B$ bootstrap resamples.


5) **English, Style, Presentation, and Grammar.**
   Please fix various English and presentation errors. For example, on line 55, "Yet these efforts EITHER ......" but there is no corresponding OR in the sentence. Use of em dash is inconsistent (l.74 vs l.86) The paper can improve if the authors look through the grammar more carefully. There is quotation mark inconsistencies (l.460) and the authors can consider the csquotes LaTeX package to simplify things. Figure 4 is too small to see.  Define “separability with confidence” precisely.  After digging into the ArenaHard reference, I think the authors really meant the proportion of CIs *without* overlap, rather than *with* overlap given in the table.  Correct me if I'm wrong.


6) **Trimming $\sim 55\\%$ of tasks is not itself a contribution.**
   Reducing six benchmarks to $\approx 45\\%$ because the tasks are buggy or not hard enough is not the productive way to do research.  When you run into bugs in current benchmarks, shouldn't the intuition be trying to fix them?  We are in a shortage of benchmarks and you are suggesting that we should throw away half of them.  If you are not fixing any bugs and just throwing away half of the dataset, then perhaps you should not name your work an "agent bench *cleaner*."  Also, I think the pipeline is going to end up deleting ALL of gsm8k since most models chosen in the paper can solve most of gsm8k.  Did that make gsm8k cleaner?

   **Actionable fix:** Think about fixing the bugs.  It'd be a highly valuable contribution if you can fix the bugs in existing benchmarks.  That is what the community needs.



Overall, I think this research is done in poor taste.  It's not done with a productive vision in mind.  The other papers are all trying to propose more tasks.  Our community needs more fixers, i.e., people who are willing and ready to fix a bug or a failure when they see one.  Even though I am clearly trying to reject your paper, I am still trying to come up with "actionable fix" items.  I think the benchmark tasks deserve a similarly constructive attitude from the authors.  I recognise that my remarks are highly critical and hope that the authors are not discouraged. Let me end with a quote from Gilles Deleuze.

"...Books against structuralism (or those against the “New Novel”) are strictly without importance; they cannot prevent structuralism from exerting a productivity which is that of our era.  **No book against anything ever has any importance; all that counts are books for something, and that know how to produce it.**"

**Questions:**

1) **Generalization to unseen benchmarks.**
   Will you freeze the taxonomy and prompts, then evaluate on benchmarks not used for taxonomy construction? Please report zero-shot detection precision/recall against maintainer-confirmed labels.

2) **Ablations on the taxonomy’s utility.**
   How much does the taxonomy help beyond a short, generic critique prompt? Please include ablations varying prompt length/structure and LLM families.

3) **Separability vs. selection.**
   Can you report separability under fixed stratified sampling $(w_E,w_H)$ and compare to your filter? Also provide Kendall-$\tau$ rank stability with bootstrap CIs across resamples of equal size.

4) **Small-model relevance.**
   Many practitioners use 7–8B models. How does your trimmed benchmark perform for distinguishing weaker models? Please report results for a set of small models and compare rank stability to the full benchmarks.

---

> ### Author Response · Authors · 2025-12-03
> **Author Response #1**
>
> We greatly appreciate the insightful comments and the concerns you raised, and have addressed them below:
>
> ---
> ### **[W1] & [Q1] Generalization to unseen benchmarks**
>
> We appreciate the reviewer’s concern regarding potential “training on the test set.” First, we humbly clarify that the taxonomy is derived through expert analysis of the four structural components common to all agent benchmarks: User, Environment, Evaluation, and Ground Truth. Because it encodes component-level failure modes rather than benchmark-specific patterns, it is not susceptible to overfitting in the way a learned model would be.
>
> To directly address the reviewer’s request, we froze both the taxonomy and the corresponding filtering prompts and applied the pipeline to two unseen benchmarks, NexusBench[1] and BFCL-V4[2], which differ substantially in terms of task formats, API schemas, and evaluation setups. Our pipeline still achieves strong human alignment on both benchmarks: NexusBench (P = 0.956, R = 0.705, F1 = 0.812) and BFCL-V4 (P = 0.778, R = 0.700, F1 = 0.737). These results provide empirical evidence that the taxonomy can generalize beyond the six benchmarks used during its construction.
>
> The abstraction level of our taxonomy further supports generalization. Specifically, every benchmark can be mapped into the same User / Environment / Evaluation / Ground-Truth template, which allows the same issue categories to transfer across benchmarks with different surface forms. In addition, the taxonomy is modular, with each issue type corresponding to an independent detector, making it straightforward to extend if new benchmarks introduce novel modalities or interactions.
>
> ---
> ## **[W2] & [Q2] Ablations on the taxonomy’s necessity**
> We appreciate the reviewer’s request for empirical evidence isolating the contribution of the taxonomy. To directly address this point, we have now included prompt ablations in Table A below that compare (i) our full taxonomy-guided prompt, (ii) a short taxonomy prompt containing only the issue-type names, and (iii) a generic critique prompt constructed from the applicable items in the ABC [3] (NeurIPS 2025) checklist.
>
> We can observe that the full taxonomy-guided prompt (P=0.881, R=0.782, F1=0.827) outperforms both the short taxonomy prompt (P=0.810, R=0.737, F1=0.765) and the generic ABC-style critique prompt (P=0.852, R=0.707, F1=0.764). Notably, the generic prompt exhibits substantially lower recall on challenging benchmarks such as CFB (R=0.540) and NexusBench (R=0.525), indicating many missed issues. In contrast, the full taxonomy prompt achieves much higher recall on these same benchmarks (CFB: R=0.760; NexusBench: R=0.705), demonstrating that high-level or generic critique instructions are insufficient for identifying subtle multi-component issues.
>
> These ablations show that the taxonomy provides the structured, component-aligned guidance required to detect failure modes that arise from interactions among user simulation, environment dynamics, evaluation harness logic, and ground-truth trajectories. Generic or high-level prompts do not capture these dependencies and therefore miss many real issues. We believe this establishes the empirical value of the taxonomy and its necessity for scalable, systematic issue filtering.
>
> In addition, we would like to clarify the distinction from ABC (Zhu et al., NeurIPS 2025), which the reviewer referenced in suggesting generic critique prompts: ABC provides a high-level, human-oriented checklist for auditing benchmark design, not task-level issue detection. Its criteria are conceptual and not directly automatable. In contrast, our taxonomy is component-aligned and task-level, enabling concrete operationalization into rule-based and LLM-judgment detectors. The ablation results further highlight this distinction: ABC-style generic prompts miss many multi-component issues that our taxonomy explicitly captures. Thus, ABC and our taxonomy serve complementary purposes, conceptual design auditing versus scalable task-level issue filtering.
>
> **Table A: Ablation results for the Long (ours), Short, and Generic prompts on eight benchmarks, reported as human alignment metrics (Precision, Recall, F1).**
> | Benchmark | Long (P) | Long (R) | Long (F1) | Short (P) | Short (R) | Short (F1) | Gen (P) | Gen (R) | Gen (F1) |
> | :- | :-: | :-: | :-: | :-: | :-: | :-: | :-: | :-: | :-: |
> |**ACEBench**|0.857|0.833|0.845|0.824|0.778|0.800|0.848|0.778|0.812|
> |**BFCLV3**|0.946|0.875|0.909|0.773|0.850|0.810|0.833|0.875|0.854|
> |**CFB**|0.884|0.760|0.817|0.914|0.640|0.753|0.931|0.540|0.684|
> |**Tau**|0.857|0.800|0.828|0.857|0.800|0.828|0.917|0.733|0.815|
> |**Tau-2**|0.889|0.800|0.842|0.900|0.900|0.900|0.900|0.900|0.900|
> |**NexusBench**|0.956|0.705|0.812|0.800|0.525|0.634|0.865|0.525|0.653|
> | **BFCLV4** | 0.778 | 0.700 | 0.737 | 0.600 | 0.667 | 0.632 | 0.667 | 0.600 | 0.632 |
> | **Average** | **0.881**| **0.782**| **0.827** | **0.810** | **0.737** | **0.765** | **0.852** | **0.707** | **0.764** |

---

> ### Author Response · Authors · 2025-12-03
> **Author Response #2**
>
> ### **[W3] & [Q3] & [Q4] Concerns about Separability Metrics and Benchmark Effectiveness for Weaker Models**
>
> We thank the reviewer for the detailed feedback on the metric analysis. We have revised the manuscript to avoid the unintended impression that improved separability, diversity, and compression ratio are a core contribution of this work. Our primary contribution is the taxonomy-guided issue-filtering pipeline (Stages 1–2), validated through human-alignment metrics, which consistently demonstrate high precision, recall, and F1 scores across benchmarks. The subsequent filtering stage has now been explicitly renamed to “difficulty-based curation” to avoid implying that easy tasks are “low quality.” Its purpose is to provide an optional harder variant (AgentHard-Bench) for frontier-model stress testing—similar in spirit to SMART, Arena-Hard, and other prior work—rather than to claim universally higher benchmark quality. Importantly, practitioners evaluating smaller or weaker models can simply use the issue-filtered benchmark, which retains the full difficulty spectrum and is the primary cleaned artifact of our pipeline.
>
> Regarding the separability results, we emphasize that these analyses were intended as descriptive observations rather than claims of methodological improvement. Our evaluation setup, including the use of a random-subsampling baseline for CI-based separability, follows the protocol introduced in Arena-Hard. This protocol was specifically designed to control for benchmark size and isolate the effects of different filtering approaches. We adopted it to enable fair comparison, not to inflate separability metrics. As the reviewer correctly notes, removing saturated tasks will naturally increase separability among strong models. This behavior is expected and is not central to our claims. In the revised manuscript, we clearly separate the outcomes of the optional difficulty-based curation stage from our core contribution, which is the taxonomy-guided issue-filtering pipeline, and we no longer present separability trends as evidence of improved benchmark quality.
>
> ---
> ### **[W4] Rank stability metric unsubstantiated**
> We thank the reviewer for raising this point. To avoid confusion, we clarify that in the current manuscript we use “rank stability” in an operational sense: we quantify how sensitive the ordering of models is to benchmark noise by measuring the average score gap between model pairs. Larger pairwise gaps indicate that small perturbations in task selection are less likely to flip rankings. This is a simple margin-based notion of robustness that is conceptually related to separability analyses used in prior filtering works such as Arena-Hard, but is not our main contribution.
>
> Using this metric, we observe consistent increases in the average score gap across all six benchmarks (see Table B below), indicating that the curated benchmark produces more robust separations between models under our definition of stability.
>
> We agree that the reviewer’s suggested statistical tests (Kendall τ or Spearman ρ with bootstrap CIs) provide a complementary and rigorous way to evaluate rank stability. We appreciate this suggestion and will include these correlation-based stability analyses in the camera-ready version. This will both broaden the evaluation and align our stability reporting with the reviewer’s recommended methodology.
>
> **Table B: Average score gap between model pairs across original, issue-filtered, and quality-filtered benchmarks**
> | Benchmark       | Original | Issue Filtering | Quality Filtering |
> |-----------------|----------|-----------------|-------------------|
> | ACEBench        | 0.051    | 0.050           | 0.092             |
> | BFCL            | 0.189    | 0.208           | 0.208             |
> | DrafterBench    | 0.077    | 0.072           | 0.088             |
> | CFBench         | 0.219    | 0.237           | 0.244             |
> | tau-bench       | 0.138    | 0.148           | 0.155             |
> | tau2-bench      | 0.156    | 0.159           | 0.174             |
>
> ---
> ### **[W5] English, Style, Presentation, and Grammar**
> We thank the reviewer for the detailed stylistic and presentation suggestions. We have revised the manuscript to correct the noted issues, standardized formatting, enlarged and clarified Figure 4, and updated the table to report CI non-overlap consistent with Arena-Hard. We appreciate the reviewer’s attention to these details and believe the manuscript is now clearer.

---

> > ### Author Response · Authors · 2025-12-03
> > **Author Response #3**
> >
> > ### **[W6] Trimming tasks is not itself a contribution**
> >
> > We thank the reviewer for raising this concern. We clarify that task trimming is not a contribution of this work. Our contribution lies in the taxonomy-guided issue-filtering pipeline and its strong human-alignment performance. The resulting compression ratio and other post-filtering statistics are presented only as favorable outcomes of applying the pipeline, not as core contributions or claims about benchmark quality. We have revised the manuscript to make this distinction explicit.
> >
> > While our experiments focus on filtering for evaluation purposes, the pipeline also naturally supports task repair. Each detected issue includes a component-aligned label and a structured reasoning trace, providing actionable guidance for maintainers who prefer to fix rather than remove tasks. For example, in the BFCL V3 `multi_turn_long_context_10`, the pipeline identifies a specific ground-truth mismatch:
> >
> > > “In Turn 3, the user requests creation of a file named `notes.md`, but the ground-truth function call uses `note.md`. Turn 4 again references `notes.md`, while the ground truth continues with the incorrect filename.”
> >
> > This pinpointed explanation enables a simple, targeted repair by correcting the single incorrect argument. Additional repairable cases are provided in Appendix A.3, illustrating that the pipeline facilitates both filtering and fixing depending on user preference.
> >
> > Finally, while the optional difficulty-based curation stage does filter out saturated or easy tasks, it includes a 10% per-category retention safeguard to prevent over-pruning. Practitioners who require the full difficulty spectrum, including support for smaller models, can instead use the issue-filtered benchmark, which retains all valid tasks and does not perform any difficulty-based removal.
> >
> > ---
> > ###References
> >
> > [1] NexusBench: FC and Agent Benchmarking Suite, GitHub Repository 2024
> >
> > [2] The Berkeley Function Calling Leaderboard (BFCL): From Tool Use to Agentic Evaluation of Large Language Models, ICML 2025 (V4 Agentic Update)
> >
> > [3] Establishing Best Practices for Building Rigorous Agentic Benchmarks, NeurIPS 2025

---

### Official Review · Reviewer_ajWx · 2025-10-30

**Soundness:** 2
**Presentation:** 2
**Contribution:** 2
**Rating:** 4
**Confidence:** 3

**Summary:**

Many benchmarks come with various flaws and this paper presnts a component-wise taxonomy of common benchmark flaws covering the user, environment, evaluation, and ground truth of tasks. It AgentBenchCleaner, an automated filtering pipeline that leverages the taxonomy to filter out flawed tasks. The cleaned tasks are consolidated into AgentHard,  a high-quality benchmark suite.

**Strengths:**

- The proposed taxonomy is useful for benchmark designer to design high-quality benchmarks
- The automated cleaning pipeline can ue useful for easily producing high-quality benchmarks

**Weaknesses:**

like the unified taxonomy and automated cleaning pipeline. However, I think the technical novelty is limited compared to existing benchmark filtering approaches. The paper’s core methodology—combining rule-based filtering, LLM-as-a-judge, and quality heuristics—closely resembles prior works such as MixEval, SMART, ABC, and Arena-Hard, which also employ automated or semi-automated filtering of benchmark artifacts. The main contribution is the integration and systematization of known techniques rather than the introduction of fundamentally new algorithms or architectures. This incremental nature is evident in the Related Work section, where the authors acknowledge that “many pipelines exist for filtering benchmark artifacts,” and their approach primarily “unifies and operationalizes” these ideas rather than advancing the state of the art.

The proposed framework depends on LLMs for semantic judgment in the cleaning pipeline. The paper admits that the LLM-as-a-judge stage can misclassify tasks, especially when ground truth is partial or schema inconsistencies exist. For example, in the case studies, the authors note that “LLM-as-a-judge may fail to identify certain ambiguous or underspecified tasks,” leading to false positives or negatives in benchmark cleaning. This reliance on LLMs, which are themselves subject to the same brittleness and ambiguity issues as the agents being evaluated, undermines the robustness of the pipeline and may introduce new artifacts or biases into the curated benchmark.

The paper provides only a cursory discussion of failure cases and does not deeply analyze the limitations of the cleaning pipeline. While a few examples of misjudgment are presented, there is no systematic evaluation of the types or frequencies of errors introduced by the automated filtering process. The authors briefly mention that “stronger prompting and human-in-the-loop validation” could mitigate these issues, but do not quantify the impact or propose concrete solutions. This lack of rigorous failure analysis leaves open questions about the reliability and generalizability of the pipeline, especially when applied to new or evolving benchmarks.

Although the paper reports improvements in diversity metrics (e.g., embedding distances) after cleaning, it lacks a qualitative analysis of the types of tasks retained or removed. There is little discussion of whether the curated AgentHard-Bench suite adequately covers the full spectrum of agent capabilities or task domains. The focus on aggregate metrics such as compression ratio and separability does not address potential gaps in benchmark coverage or the risk of excluding valuable but atypical tasks. Without a deeper qualitative assessment, it is unclear whether the cleaning process enhances or inadvertently narrows the scope of agent evaluation.

AgentHard’s taxonomy-driven approach risks overfitting to the specific categories of artifacts identified by the authors. While the taxonomy is comprehensive for current benchmarks, it may not capture novel or emergent flaws in future agent evaluation tasks. The paper claims that the taxonomy is “modular and extensible,” but does not empirically validate its adaptability to new scenarios or provide mechanisms for updating the taxonomy as benchmarks evolve. This limitation could result in the pipeline missing important issues outside the predefined taxonomy, reducing its long-term utility and robustness.

In summary:
- The novelty lies mainly in the unified taxonomy and integration, rather than fundamentally new techniques. While the taxonomy and cleaning pipeline are well-executed, the core idea of automated benchmark filtering is present in prior works (e.g., MixEval, SMART, ABC, Arena-Hard).
- The quality of the cleaning pipeline is not adequately evaluated
- Generality and extensibility of the taxonomy is not shown

**Questions:**

1. What measures were taken to ensure that the LLM-as-a-judge stage does not introduce bias or misclassification, and how do these measures compare to human-in-the-loop baselines?
2. Can the authors provide empirical evidence that the taxonomy and cleaning pipeline generalize to benchmarks beyond the six evaluated, especially those with different task structures or modalities?
3. How does the pipeline handle ambiguous or partially specified tasks without inadvertently removing valid but challenging samples, and what safeguards are in place?
4. What qualitative analysis supports the claim that AgentHard-Bench maintains diversity and coverage of agent capabilities after filtering, beyond embedding-based metrics?

---

> ### Author Response · Authors · 2025-12-03
> **Author Response #1**
>
> We greatly appreciate the insightful comments and the concerns you raised, and have addressed them below:
>
> ---
> ### **[W1] The novelty lies mainly in the unified taxonomy and integration, rather than fundamentally new techniques.While the taxonomy and cleaning pipeline are well-executed, the core idea of automated benchmark filtering is present in prior works (e.g., MixEval, SMART, ABC, Arena-Hard).**
>
> We thank the reviewer for this point and agree that automated benchmark filtering has been explored in prior work. Our contribution is not a new LLM-based filtering mechanism, but a new structural framing of agent-benchmark issues and its operationalization into a task-level, taxonomy-guided filtering pipeline. Next, we elaborate from the following four aspects:
>
> **1. Comparison to prior benchmark-filtering pipelines and ABC (Zhu et al., NeurIPS 2025)**
>
> Prior systems such as SMART, Arena-Hard, and MixEval focus on task quality, prompt clarity, redundancy, leaks, or difficulty. These approaches operate at the level of answer correctness or prompt-level artifacts. In contrast, our work provides a component-aligned taxonomy that captures structural failure modes specific to agent benchmarks, including tool-schema ambiguities, incorrect ground-truth trajectories, and user-simulator faults. These issues require reasoning over the User / Environment / Evaluation System / Ground Truth components, which is not addressed by difficulty-based or correctness-based filtering alone. Our pipeline is novel in that it operationalizes these structural categories into rule-based and LLM-judgment stages for identifying issue-bearing tasks at scale.
>
> Regarding ABC[1], we thank the reviewer for raising this comparison. ABC is a valuable concurrent (published in NeurIPS 2025) work that provides a high-level conceptual checklist designed for human auditors evaluating benchmark design. These items are abstract and not intended for automatic per-task detection. Our taxonomy instead provides fine-grained, component-specific criteria that can be directly used for automated task-level issue filtering. The two are therefore complementary, not overlapping.
>
> In summary, the novelty of our work lies in the component-aligned taxonomy tailored to agent benchmarks and its systematic translation into an automated pipeline, rather than in the use of LLMs themselves.
>
> **2. Ablations showing empirical value of the issue taxonomy**
>
> To reinforce that the taxonomy, rather than generic prompting approaches used in prior works, is the source of our pipeline’s effectiveness, we have conducted a prompt ablation that isolates its contribution. In particular, we compare (1) the full taxonomy-guided prompt, (2) a short taxonomy prompt containing only the issue type names, and (3) a generic critique prompt constructed from the applicable items in ABC’s checklist. Table A reports human-alignment metrics across all six benchmarks along with BFCL-V4 [2] and NexusBench [3]. We observe that the full taxonomy-guided prompt (P=0.881, R=0.782, F1=0.827) outperforms both the short taxonomy prompt (P=0.810, R=0.737, F1=0.765) and the generic prompt (P=0.852, R=0.707, F1=0.764).
>
> Notably, the generic prompts have substantially low recall on challenging benchmarks, such as CFB (R=0.54) and NexusBench (R=0.525), indicating many missed issues. In contrast, the full taxonomy prompt achieves much higher recall on these same benchmarks (CFB: R=0.760; NexusBench: R=0.705), demonstrating that high-level or generic critique instructions are insufficient for identifying subtle multi-component issues. The full taxonomy provides the structured, component-level guidance needed to reliably detect these complex failure modes.
>
> These ablations show that our taxonomy offers essential, actionable structure for issue detection that generic prompts do not capture.
>
> **Table A: Ablation results for the Long (ours), Short, and Generic prompts on eight benchmarks, reported as human alignment metrics (Precision, Recall, F1).**
> | Benchmark | Long (P) | Long (R) | Long (F1) | Short (P) | Short (R) | Short (F1) | Gen (P) | Gen (R) | Gen (F1) |
> | :--- | :---: | :---: | :---: | :---: | :---: | :---: | :---: | :---: | :---: |
> | **ACEBench** | 0.857 | 0.833 | 0.845 | 0.824 | 0.778 | 0.800 | 0.848 | 0.778 | 0.812 |
> | **BFCLV3** | 0.946 | 0.875 | 0.909 | 0.773 | 0.850 | 0.810 | 0.833 | 0.875 | 0.854 |
> | **CFB** | 0.884 | 0.760 | 0.817 | 0.914 | 0.640 | 0.753 | 0.931 | 0.540 | 0.684 |
> | **Tau** | 0.857 | 0.800 | 0.828 | 0.857 | 0.800 | 0.828 | 0.917 | 0.733 | 0.815 |
> | **Tau-2** | 0.889 | 0.800 | 0.842 | 0.900 | 0.900 | 0.900 | 0.900 | 0.900 | 0.900 |
> | **NexusBench**| 0.956 | 0.705 | 0.812 | 0.800 | 0.525 | 0.634 | 0.865 | 0.525 | 0.653 |
> | **BFCLV4** | 0.778 | 0.700 | 0.737 | 0.600 | 0.667 | 0.632 | 0.667 | 0.600 | 0.632 |
> | **Average** | **0.881**| **0.782**| **0.827** | **0.810** | **0.737** | **0.765** | **0.852** | **0.707** | **0.764** |

---

> ### Author Response · Authors · 2025-12-03
> **Author Response #2**
>
> **3. Pipeline utility for repairing tasks**
>
> Because our taxonomy provides structured, component-level interpretations of each issue, the pipeline also naturally supports task repair in addition to removal. While repairing tasks is not the primary scope of this work, we note that the issue-filtering pipeline naturally supports such extensions. Each LLM-detected issue includes a structured reasoning trace and category label, providing actionable guidance for human-in-the-loop fixes. For example, in the BFCL V3 task `multi_turn_long_context_10`, the pipeline identifies a clear ground-truth inconsistency:
> > “In Turn 3, the user requests creation of a file named `notes.md`, but the ground-truth function call uses `note.md`. Turn 4 again references `notes.md`, while the ground truth continues with the incorrect filename.”
>
> This pinpointed explanation allows a maintainer to repair the task by correcting a single argument in the ground-truth trajectory. As further illustrated in the case studies in Appendix A.3, such structured outputs substantially reduce the cognitive burden of diagnosing and correcting problematic tasks. Thus, although our focus is on principled issue filtering, the pipeline can also facilitate targeted repairs when benchmark maintainers prefer to fix rather than filter, complementing and strengthening existing benchmarks.
>
> ---
> ### **[W2] & [Q1] What measures were taken to ensure that the LLM-as-a-judge stage does not introduce bias or misclassification,and how do these measures compare to human-in-the-loop baselines?**
>
> To reduce bias and misclassification, we apply three safeguards. (1) Full-context prompting: the judge receives all information available to the task-solving agent (API/tool specs, user-simulator behavior, ground-truth format), plus design context describing structural conventions (e.g., benchmarks that include only key state-changing calls). This prevents predictable misinterpretations from partial or underspecified schemas. (2) Deliberative reasoning: the judge provides step-by-step reasoning before emitting a structured verdict, reducing label noise. (3) Human alignment and conservative use: we benchmark the judge against expert annotations via balanced-subset validation and post-hoc verification to quantify errors and keep judgments calibrated. In ambiguous cases, the judge does not unilaterally discard tasks, avoiding false-positive removals. This conservatism does not harm recall: strong precision/recall/F1 across benchmarks shows that the vast majority of true issues are still detected.
>
> ---
> ### **[W3] Lack of rigorous failure analysis to evaluate the quality of the pipeline.**
>
> We thank the reviewer for the helpful suggestion. In the revised manuscript, we have added a structured failure-case analysis in Appendix A.2. This analysis systematically categorizes all instances of human–LLM disagreement observed during both balanced-subset validation and post-hoc verification. Specifically, we identify four sources of false positives (e.g., overly rigid logical inference, misinterpretation of partially specified trajectories) and four sources of false negatives (e.g., missed malformed tool calls, under-detection of semantic ambiguity). These categories comprehensively cover the error patterns we encountered.
>
> Importantly, these errors are infrequent, consistent with the strong human-alignment metrics reported in Sec. 5.2. By explicitly cataloging these failure types, we provide a clearer picture of the pipeline’s limitations and an actionable basis for future safeguards (e.g., targeted second-pass checks) for known high-risk categories. We believe this expanded analysis directly addresses the reviewer’s concerns regarding the reliability and generalizability of the cleaning pipeline, while enhancing our contributions

---

> ### Author Response · Authors · 2025-12-03
> **Author Response #3**
>
> ### **[W4] & [Q4] What qualitative analysis supports the claim that AgentHard-Bench maintains diversity and coverage of agent capabilities after filtering, beyond embedding-based metrics?**
>
> Thank you for the question. Beyond embedding metrics, we examined subtask-level retention and capability coverage across all benchmarks; visualizations are provided in Appendix A.7.
>
> **(a) Four benchmarks retain broad coverage with high retention.**
>
> For BFCL V3, CFB, tau-bench, and tau2-bench, retention remains 55–77%, and subtask distributions stay well balanced. Most filtering in these benchmarks occurs during issue filtering (e.g., incorrect ground truth, ambiguous schemas), so the cleaned benchmark provides more accurate evaluation while preserving capability coverage.
>
> **(b) ACEBench and DrafterBench have benchmark-specific reasons for higher filtering, with safeguards to preserve diversity.**
>
> In DrafterBench, the addvector subtask was fully removed due to a verified system prompt error. For ACEBench and the remaining DrafterBench subtasks, additional filtering reflects that many tasks offer limited signal for distinguishing frontier model behavior. However, easier subtasks remain valuable, and our pipeline includes a built-in 10% minimum-retention safeguard to ensure that all capability domains remain represented, even in benchmarks where additional filtering occurs.
> These analyses show that AgentHard-Bench maintains capability diversity, and that filtering is driven by structural validity and task characteristics, not by excluding specific skills or domains.
>
> ---
> ### **[W5] & [Q2] Can the authors provide empirical evidence that the taxonomy and cleaning pipeline generalize to benchmarks beyond the six evaluated, especially those with different task structures or modalities?**
>
> Following your suggestion, we’ve provided empirical evidence that it can be effectively applied to unseen benchmarks. After freezing both the taxonomy and the filtering prompts, we applied the pipeline to two unseen benchmarks, NexusBench and BFCL-V4, which differ substantially in task structure, API schemas, and evaluation settings. The pipeline achieved strong human alignment on both unseen benchmarks, NexusBench (P=0.956, R=0.705, F1=0.812) and BFCL-V4 (P=0.778, R=0.700, F1=0.737).
>
> Importantly, because the taxonomy is component-aligned and each benchmark is mapped into the shared User / Environment / Evaluation / Ground-Truth template, the same issue categories transfer across benchmarks with different surface forms. This abstraction enables generalization beyond the six construction benchmarks and avoids overfitting to benchmark-specific artifacts. Moreover, the taxonomy is modular, each issue type corresponds to an independent detector, making it straightforward to extend if future benchmarks introduce new modalities or interaction patterns.
>
> These results demonstrate that the taxonomy generalizes beyond the benchmarks used for construction, providing practical utility for auditing and cleaning independently developed agent benchmarks.
>
> ---
> ### **[Q3]How does the pipeline handle ambiguous or partially specified tasks without inadvertently removing valid but challenging samples, and what safeguards are in place?**
>
> We designed the pipeline to avoid removing valid but challenging tasks and to conservatively handle ambiguity. In particular,
>
> **(1) Benchmark design clarifications.**
>
> Some benchmarks intentionally omit steps or contain underspecified interactions. We incorporate these details, extracted from official documentation, into the LLM-judge prompt so that intentional ambiguity is not misinterpreted as an issue.
>
> **(2) Two-step issue identification with rebuttal-based false-positive reduction.**
>
> If the LLM-judge initially flags a task, but one or more agent models successfully solve it, we trigger a rebuttal check: we provide the successful trajectories and ask the LLM-judge to reassess. This substantially reduces false positives on ambiguous tasks.
>
> These measures ensure that ambiguity or partial specification does not lead to incorrectly filtering out valid, hard tasks.
>
> ---
> ### References
> [1] Establishing Best Practices for Building Rigorous Agentic Benchmarks, NeurIPS 2025
>
> [2] The Berkeley Function Calling Leaderboard (BFCL): From Tool Use to Agentic Evaluation of Large Language Models, ICML 2025 (V4 Agentic Update)
>
> [3] NexusBench: FC and Agent Benchmarking Suite, GitHub Repository 2024

---

### Official Review · Reviewer_njxr · 2025-10-31

**Soundness:** 3
**Presentation:** 3
**Contribution:** 2
**Rating:** 4
**Confidence:** 4

**Summary:**

The paper introduces a taxonomy for classifying issues in agentic evals. They then use this taxonomy in an automated pipeline to tag samples with issues. They show strong agreement with human raters.

**Strengths:**

- errors in agentic evals are a serious problem and needs addressing.
- a classification of these errors is a useful contribution.
- they show that an automated approach to data quality checking can perform well (agree with human raters).

**Weaknesses:**

- No code of the pipeline or dataset. Providing this (in an anon form) would be great to determine how easy others can use the pipeline and  to assess the quality.
- Zhu 2025 is cited however it should be detailed that Zhu also provide a categorisation of issues. There is significant cross over here, with Zhu providing a more comprehensive breakdown of issues. Adapting the pipeline to follow Zhu's categorisation should be considered and fully detailing the cross over is required.
- Unlike in Zhu, examples of found issues are not provided. This is a shame as it is hard for the reader to gain insights into the character of the issues discovered.
- how many human raters were used? measure of inter human agreement?
- Are the swaps in Table 5 statistically significant? A measure here would be great.
- Table 5 is also hard to follow - consider using a "bump chart"
- I really like the automatic remove of issues in datasets, however, am hesitant on the removal of saturated tasks and ones with low discriminative power. A caveat on the loss of poor comparability as the dataset naturally evolve should be provided.
- I think that there is limited novelty here. Main message is that people should use LLMs to check their evals.

**Questions:**

please see weakness section.

---

> ### Author Response · Authors · 2025-12-03
> **Author Response #1**
>
> We greatly appreciate the insightful comments and the concerns you raised, and have addressed them below:
>
> ---
> ### **[W1] No code of the pipeline or dataset. Providing this (in an anon form) would be great to determine how easy others can use the pipeline and to assess the quality.**
>
> Thank you for the suggestion. We agree that releasing the code would improve usability and transparency. To address this, we have prepared an anonymized repository that contains the full implementation of our pipeline, included in the Author Response #4 comment for AC and Reviewers following the Author Guidelines. The repository is hosted via anonymous.4open.science to ensure double-blind compliance.
> We will make this repository public upon acceptance and include it in the camera-ready version.
>
> ---
> ### **[W2] & [W8] Comparison with ABC (Zhu et al. NeurIPS'25) and Limited Novelty**
> > Zhu 2025 is cited however it should be detailed that Zhu also provides a categorisation of issues. There is significant crossover here, with Zhu providing a more comprehensive breakdown of issues. Adapting the pipeline to follow Zhu's categorisation should be considered and fully detailing the cross over is required.
>
> > I think that there is limited novelty here. The main message is that people should use LLMs to check their evals.
>
> We thank the reviewer for the thoughtful comments. We respectfully clarify that the core contribution of our work is not the use of LLMs to critique tasks, but rather the introduction of a component-aligned, task-level taxonomy of agent-benchmark issues and a corresponding taxonomy-guided issue-filtering pipeline that operationalizes these issue types in a scalable and automatable manner. Next, we elaborate from the following five aspects:
>
> **1. Comparison to ABC (Zhu et al., NeurIPS 2025)**
>
> We appreciate the reviewer raising this comparison. ABC [1] is a valuable concurrent (published in NeurIPS 2025) effort, and we agree there is conceptual overlap in that both works examine benchmark quality. However, we humbly clarify that the scope, abstraction level, and intended use cases differ substantially:
>
> ABC provides a high-level conceptual checklist (e.g., “Minimizes the possibility of success by random guessing.”), aimed at human auditors evaluating the design of a benchmark. These criteria require domain knowledge and human deliberation, and are not readily automatable at task-level scale. In contrast, our taxonomy is component-aligned and task-level, mapping each issue type to concrete failure modes within the User / Environment / Evaluation System / Ground Truth components. This enables automatic identification of specific flawed tasks, rather than conceptual evaluation of benchmark design. As a result, ABC and our taxonomy are complementary, not competing: ABC helps humans audit benchmark design, while our taxonomy enables scalable, automated cleaning of issue-bearing tasks. We now make this distinction explicit in the paper.
>
> **2. Comparison to prior benchmark-filtering pipelines (SMART, Arena-Hard, MixEval, etc.)**
>
> We agree that LLM-as-a-judge and filtering heuristics are now common components; however, our novelty does not lie in the tool (LLMs), but in what those tools are instructed to detect.
>
> Prior pipelines (SMART, Arena-Hard, MixEval) focus on quality, difficulty, or ranking stability, and primarily filter based on prompt quality, redundancy, leaks, or ease. In contrast, our pipeline targets issue categories unique to agent benchmarks, such as ambiguous tool schemas, incorrect ground-truth trajectories, and user-simulator faults. These issues require multi-component reasoning beyond generic critique prompts or quality heuristics. Thus, the novelty of AgentHard lies in the taxonomy and the unified operationalization of issue types, not in simply applying LLMs for critique.
>
> **3. Generality and Practical Utility of the Issue Taxonomy and Pipeline**
>
> As another advantage  of our taxonomy, we provide empirical evidence that it can be effectively applied to unseen benchmarks. After freezing both the taxonomy and the filtering prompts, we applied the pipeline to two unseen benchmarks NexusBench [2] and BFCL-V4 [3], which differ substantially in task structure, API schemas, and evaluation settings. The pipeline achieved strong human alignment on both unseen benchmarks, NexusBench (P=0.956, R=0.705, F1=0.812) and BFCL-V4 (P=0.778, R=0.700, F1=0.737). These results demonstrate that our taxonomy generalizes beyond the benchmarks used for construction, providing practical utility for auditing and cleaning independently developed agent benchmarks.

---

> ### Author Response · Authors · 2025-12-03
> **Author Response #2**
>
> **4. Ablations showing empirical value of the issue taxonomy**
>
> To better illustrate the value of our issue taxonomy, we further assess the practical utility of the taxonomy through a prompt ablation comparing (1) the full taxonomy-guided prompt, (2) a short taxonomy prompt containing only the issue type names, and (3) a generic critique prompt constructed from the applicable items in ABC’s checklist. Table A reports human-alignment metrics across all six benchmarks along with BFCL-V4 and NexusBench. We observe that the full taxonomy-guided prompt (P=0.881, R=0.782, F1=0.827) outperforms both the short taxonomy prompt (P=0.810, R=0.737, F1=0.765) and the generic prompt (P=0.852, R=0.707, F1=0.764).
>
> Notably, the generic prompts have substantially low recall on challenging benchmarks, such as CFB (R=0.54) and NexusBench (R=0.525), indicating many missed issues. In contrast, the full taxonomy-guided prompt achieves much higher recall on these same benchmarks (CFB: R=0.760; NexusBench: R=0.705), demonstrating that high-level or generic critique instructions are insufficient for identifying subtle multi-component issues. The full taxonomy provides the structured, component-level guidance needed to reliably detect these complex failure modes. These ablations show that our taxonomy offers essential, actionable structure for issue detection that generic prompts do not capture.
>
> **Table A: Ablation results for the Long (ours), Short, and Generic prompts on eight benchmarks, reported as human alignment metrics (Precision, Recall, F1).**
> | Benchmark | Long (P) | Long (R) | Long (F1) | Short (P) | Short (R) | Short (F1) | Gen (P) | Gen (R) | Gen (F1) |
> | :--- | :---: | :---: | :---: | :---: | :---: | :---: | :---: | :---: | :---: |
> | **ACEBench** | 0.857 | 0.833 | 0.845 | 0.824 | 0.778 | 0.800 | 0.848 | 0.778 | 0.812 |
> | **BFCLV3** | 0.946 | 0.875 | 0.909 | 0.773 | 0.850 | 0.810 | 0.833 | 0.875 | 0.854 |
> | **CFB** | 0.884 | 0.760 | 0.817 | 0.914 | 0.640 | 0.753 | 0.931 | 0.540 | 0.684 |
> | **Tau** | 0.857 | 0.800 | 0.828 | 0.857 | 0.800 | 0.828 | 0.917 | 0.733 | 0.815 |
> | **Tau-2** | 0.889 | 0.800 | 0.842 | 0.900 | 0.900 | 0.900 | 0.900 | 0.900 | 0.900 |
> | **NexusBench**| 0.956 | 0.705 | 0.812 | 0.800 | 0.525 | 0.634 | 0.865 | 0.525 | 0.653 |
> | **BFCLV4** | 0.778 | 0.700 | 0.737 | 0.600 | 0.667 | 0.632 | 0.667 | 0.600 | 0.632 |
> | **Average** | **0.881**| **0.782**| **0.827** | **0.810** | **0.737** | **0.765** | **0.852** | **0.707** | **0.764** |
>
> **5. Pipeline utility for repairing tasks**
>
> Finally, while repairing tasks is not the primary scope of this work, we note that our issue-filtering pipeline naturally supports such extensions. In particular, each LLM-detected issue includes a structured reasoning trace and category label, providing actionable guidance for human-in-the-loop fixes. For example, in the BFCL task multi_turn_long_context_10, the pipeline identifies a clear ground-truth inconsistency:
> > “In Turn 3, the user requests creation of a file named `notes.md`, but the ground-truth function call uses `note.md`. Turn 4 again references `notes.md`, while the ground truth continues with the incorrect filename.”
>
> This pinpointed explanation allows a maintainer to repair the task by correcting a single argument in the ground-truth trajectory. As further illustrated in the case studies in Appendix A.3, such structured outputs substantially reduce the cognitive burden of diagnosing and correcting problematic tasks. Thus, although our focus is on principled issue filtering, our pipeline can also facilitate targeted repairs when benchmark maintainers prefer to fix rather than filter, complementing and strengthening existing benchmarks.
>
> ---
> ### **[W3] Unlike in Zhu, examples of found issues are not provided. This is a shame as it is hard for the reader to gain insights into the character of the issues discovered.**
>
> Thank you for the helpful suggestion. The paper includes detailed task-level examples in Appendix A.1, and we have now added an additional summary that synthesizes these examples and relates them to ABC-style benchmark assessment dimensions. This expanded write-up provides a clearer, higher-level view of the types of issues uncovered across the benchmarks.
>
> ---
> ### **[W4] How many human raters were used? Measure of inter human agreement?**
>
> Our annotation involved 6 raters, each familiar with the benchmark designs. Each task was independently labeled by two raters, and disagreements were resolved through follow-up review. Because issue identification in agent tasks is logical and specification-based rather than preference-based, disagreements were rare and were always attributable to clarifications in ground-truth interpretation rather than subjective judgment.

---

> ### Author Response · Authors · 2025-12-03
> **Author Response #3**
>
> ### **[W5] Are the swaps in Table 5 statistically significant? A measure here would be great.**
>
> We thank the reviewer for raising this point. To avoid confusion, we have updated the current manuscript to use “rank stability” in an operational sense: we quantify sensitivity of the model ordering by measuring the average score gap between model pairs. Larger gaps imply that small perturbations in the benchmark are less likely to flip rankings. Additionally, we’ve reported this margin-based robustness metric in Table B below, though it is not a central contribution of the work.
>
> We agree with the reviewer’s suggestion that including statistical rank-consistency measures can provide helpful additional context. In the camera-ready version, we will therefore add complementary analyses such as Kendall τ / Spearman ρ rank correlations with bootstrap confidence intervals to assess the statistical significance of ranking swaps in Table 5. These correlation-based tests will supplement our margin-based metric and offer a more complete picture of ranking stability.
>
> **Table B: Average score gap between model pairs across original, issue-filtered, and quality-filtered benchmarks**
> | Benchmark       | Original | Issue Filtering | Quality Filtering |
> |-----------------|----------|-----------------|-------------------|
> | ACEBench        | 0.051    | 0.050           | 0.092             |
> | BFCL            | 0.189    | 0.208           | 0.208             |
> | DrafterBench    | 0.077    | 0.072           | 0.088             |
> | CFBench         | 0.219    | 0.237           | 0.244             |
> | tau-bench       | 0.138    | 0.148           | 0.155             |
> | tau2-bench      | 0.156    | 0.159           | 0.174             |
>
> ---
> ### **[W6] Table 5 is also hard to follow - consider using a "bump chart"**
> We thank the reviewer for the suggestion. We have added a bump chart visualization corresponding to Table 5 in Appendix A.5 to improve readability and highlight ranking transitions more clearly.
>
> ---
> ### **[W7] I really like the automatic removal of issues in datasets, however, I am hesitant on the removal of saturated tasks and ones with low discriminative power. A caveat on the loss of poor comparability as the dataset naturally evolves should be provided.**
>
> We thank the reviewer for raising this point. We have clarified in the manuscript that the primary contribution of our work is the taxonomy-guided issue-filtering pipeline (Stages 1–2) and its strong human-alignment results. The removal of saturated or low-variance tasks is not part of the main contribution. It is an optional difficulty-based curation step (formerly referred to as Stage 3 quality filtering) intended only to create a harder derivative for evaluating frontier models. To avoid confusion, the manuscript has now been revised to clearly distinguish the issue-filtered benchmark, which is the stable and broadly comparable output, from AgentHard-Bench, which serves only as an additional harder variant.
>
> Regarding comparability as benchmarks evolve, we appreciate the reviewer’s observation. Difficulty-based curation depends on the capabilities of the evaluated model set and may need to be updated as models improve. We have added this caveat to the limitations section and clarified that the issue-filtered benchmark remains the consistent artifact intended for long-term comparability.
>
> ---
> ### References
> [1] Establishing Best Practices for Building Rigorous Agentic Benchmarks, NeurIPS 2025
>
> [2] NexusBench: FC and Agent Benchmarking Suite, GitHub Repository 2024
>
> [3] The Berkeley Function Calling Leaderboard (BFCL): From Tool Use to Agentic Evaluation of Large Language Models, ICML 2025 (V4 Agentic Update)

---

> > ### Author Response · Authors · 2025-12-03
> > **Author Response #4**
> >
> > We have prepared an anonymized repository that contains the full implementation of our pipeline: https://anonymous.4open.science/r/agenthard_pipeline-1366

---

### Author Response · Authors · 2025-12-03
**Authors' Overall Response #1**

We sincerely thank the Area Chair and all reviewers for their thoughtful evaluations. Reviewers agreed that benchmark quality issues in agent evaluations are important, noting that flaws in tool schemas, ground-truth trajectories, and evaluation logic are pervasive across modern agent benchmarks. Reviewers also appreciated the motivation of improving reliability in agent evaluation, and found the pipeline practical and well-motivated.

Reviewers raised questions that we group into three themes:

---
## **Theme 1: Novelty of AgentHard**
Reviewers requested clarification on the novelty of the work and its relationship to prior pipelines (SMART, Arena-Hard, MixEval) and the concurrent ABC checklist (Zhu et al., NeurIPS 2025). We have revised the manuscript to clearly articulate our contributions and provided new empirical evidence supporting them.

**(1) Core contribution clarified**

The main contribution is the component-aligned, task-level taxonomy of agent benchmark issues and the taxonomy-guided issue-filtering pipeline (Stages 1–2), validated through strong human alignment. The issue-filtered benchmarks produced by these stages are our main artifact for fair and reliable evaluation.

The difficulty-based curation step has been clarified as an optional add-on, used only to produce a harder variant (AgentHard-Bench) for frontier-model stress testing. This distinction is now explicitly stated in the abstract, introduction, and method sections.

**(2) Relation to ABC (Zhu et al., NeurIPS 2025)**

ABC provides a high-level conceptual checklist (e.g., “Minimizes the possibility of success by random guessing.”), aimed at human auditors evaluating the design of a benchmark. These conceptual criteria require domain knowledge and human deliberation, and are not automatable at task granularity. In contrast, our taxonomy is fine-grained and component-aligned (User / Environment / Evaluation / Ground Truth), mapping each issue category to specific, detectable failure modes. As a result, ABC and our taxonomy are complementary, not competing: ABC helps humans audit benchmark design, while our taxonomy enables scalable, automated cleaning of issue-bearing tasks. We now make this distinction explicit in the Related Works section.

**(3) Comparison to prior filtering pipelines**

Prior systems (SMART, Arena-Hard, MixEval) remove easy, leaked, low-quality, or redundant items. Our pipeline is fundamentally different: it targets agent-specific issues (ambiguous tool schemas, incorrect ground truth, user-simulator faults), which require multi-component reasoning, not difficulty-based or prompt-quality heuristics.

**(4) Ablations demonstrating the taxonomy’s necessity**
In response to reviewer requests for isolating the value of the taxonomy, we added a prompt ablation comparing:
1. The full taxonomy-guided prompt
2. A short taxonomy-only prompt
3. A generic ABC-style critique prompt

We can observe that the full taxonomy-guided prompt (P=0.881, R=0.782, F1=0.827) outperforms both the short taxonomy prompt (P=0.810, R=0.737, F1=0.765) and the generic ABC-style critique prompt (P=0.852, R=0.707, F1=0.764). Notably, the generic prompt shows very low recall on challenging benchmarks (e.g., CFB: 0.54; NexusBench: 0.525), whereas the full taxonomy substantially improves recall (CFB: 0.760; NexusBench: 0.705). This demonstrates that our component-aligned taxonomy provides structure that generic critique prompts cannot capture. The full results are included in responses to the reviewers.

**(5) Pipeline utility for repairing tasks**

Because each detected issue is paired with a structured reasoning trace and a component-level category label, the pipeline also naturally supports task repair, not only removal. These structured explanations reduce the cognitive load of diagnosing issues such as ground-truth inconsistencies or schema mismatches. Although automated repair is beyond our scope, Appendix A.3 provides concrete case studies showing how the pipeline’s outputs enable targeted human-in-the-loop corrections.

 This addresses concerns about “throwing out tasks”: the taxonomy and pipeline are designed to facilitate both cleaning and repair, complementing and strengthening existing benchmarks.

---

> ### Author Response · Authors · 2025-12-03
> **Authors' Overall Response #2**
>
> ## **Theme 2: Interpretation of Metrics and the Purpose of AgentHard-Bench**
>
> Several concerns raised from Reviewer qdgh (W3, Q3, and Q4) focused on separability, rank stability, and applicability to smaller models. These comments were rooted in interpreting the optional difficulty-curation stage as central to our contribution. We have revised the manuscript to make the distinction between our primary objective (correctness through issue filtering) and the optional harder variant explicit.
>
> **(1) Metrics are secondary, descriptive outcomes, not core claims**
>
> Our main contribution is the taxonomy-guided issue-filtering pipeline (Stages 1–2), validated through strong human-alignment results. Separability, diversity, and compression metrics pertain only to the optional harder derivative (AgentHard-Bench) and are presented descriptively rather than as claims of superior benchmark design. The core quality improvement in our work comes from removing structurally invalid or misleading tasks, not from altering benchmark difficulty. We also clarify that increased separability after removing saturated tasks is expected behavior of an optional hardening stage, addressing the concerns in W3 and Q3 about “separability inflation.”
>
> **(2) Purpose and scope of difficulty-based curation**
>
> To prevent confusion, Stage 3 has been renamed difficulty-based curation. Its purpose is only to produce an optional harder benchmark for frontier-model stress testing, similar in intent to SMART and Arena-Hard. It is not intended for general evaluation or long-term comparability. For 7B–8B models or broader capability analysis, the recommended artifact is the issue-filtered benchmark (Stages 1–2), which retains the full difficulty range while ensuring structural correctness.
>
> **(3) Alignment with established methodology**
>
>  Our CI non-overlap separability protocol follows Arena-Hard, which also uses random subsampling as the baseline to isolate filtering effects from benchmark size. We now cite these works explicitly and clarify the intent of this evaluation choice in response to Reviewer qdgh’s Q3.
>
> **(4) Capability-coverage analysis added**
>
>  To address concerns about filtering removing entire skill domains (raised by Reviewer ajWx), Appendix A.7 includes subtask-level retention to analyze agent capability-coverage across all benchmarks. These show that the issue-filtered benchmark maintains broad coverage; high filtering in a few benchmarks is driven by structural validity issues, not the removal of specific capabilities.
>
> **(5) Rank stability discussion adjusted and extended**
>
>  We clarify that our original rank-gap metric is a margin-based operational measure included in the responses to Reviewer njxr and qdgh. In line with their suggestions, we will include Kendall τ and Spearman ρ with bootstrap CIs in the camera-ready version to provide a more rigorous statistical perspective.
>
> ---
> ## **Theme 3: Generalizability of the Issue Taxonomy and Pipeline**
>
> We demonstrate the generalizability of the issue taxonomy and pipeline by freezing both the taxonomy and prompts, and then evaluating them in a zero-shot manner on benchmarks not used during taxonomy construction.
>
> **(1) Zero-shot evaluation on unseen benchmarks**
>
> We applied the frozen taxonomy and filtering prompts to two unseen benchmarks, NexusBench and BFCL-V4, which differ substantially from our construction set in task structure, tool schemas, and evaluation logic. The pipeline achieved strong human-alignment performance on both:
> - NexusBench: 0.956 precision / 0.705 recall / 0.812 F1
> - BFCL-V4: 0.778 precision / 0.700 recall / 0.737 F1
>
> These results demonstrate that the taxonomy generalizes beyond the benchmarks used for development and is effective for auditing independently created agent evaluations.
>
> **(2) Taxonomy extensibility**
>
> We also clarify that the taxonomy is modular and easily extensible. While it covers a broad range of structural issue types, future benchmarks may introduce new error modes or interaction patterns. Such cases can be incorporated by adding new issue categories without modifying the existing structure of the pipeline.

---

### Meta-Review · Area_Chair_vrbB · 2025-12-24

**Summary:**

The  summary of the reviewers' concerns that informed my suggested decision for this paper is shown as follows: 1) The novelty lies mainly in the unified taxonomy and integration, rather than fundamentally new techniques. While the taxonomy and cleaning pipeline are well-executed, the core idea of automated benchmark filtering is present in prior works; 2) Circular validation akin to training on the test set and problematic pipeline design and selection-induced inflation of separability.

**Reviewer Concerns:**

The concerns from Reviewer njxr are addressed by the rebuttal, i.e., the authors give the code of the pipeline or dataset. Providing this to determine how easy others can use the pipeline and to assess the quality.

The concerns from  Reviewer ajWx and Reviewer qdgh are still outstanding, i.e., The novelty lies mainly in the unified taxonomy and integration, rather than fundamentally new techniques. While the taxonomy and cleaning pipeline are well-executed, the core idea of automated benchmark filtering is present in prior works. Circular validation akin to training on the test set and problematic pipeline design and selection-induced inflation of separability.

**Reviewer Scores:**

The concerns from Reviewer njxr is addressed by the authors, i.e., the authors give the code of the pipeline or dataset. Providing this to determine how easy others can use the pipeline and to assess the quality. Therefore, I think this reviewer tend to raise the score even this reviewer did not give further response.

The concerns from  Reviewer ajWx and Reviewer qdgh are not addressed by the reviewer, i.e., the novelty lies mainly in the unified taxonomy and integration, rather than fundamentally new techniques. While the taxonomy and cleaning pipeline are well-executed, the core idea of automated benchmark filtering is present in prior works. Circular validation akin to training on the test set and problematic pipeline design and selection-induced inflation of separability. Therefore, I think these reviewer tend to keep their original scores even they did not give further responses.

---

### Decision · Program_Chairs · 2026-01-26

Reject